# Exploring femtosecond laser ablation in single particle aerosol mass spectrometry

Ramakrishna Ramisetty[1], Ahmed Abdelmonem[1], Xiaoli Shen[1], Harald Saathoff[1], Thomas Leisner[1] and Claudia Mohr[1, 2*]

[1]Institute of Meteorology and Climate Research, Karlsruhe Institute of Technology, Germany
[2]Department of Environmental Science and Analytical Chemistry, Stockholm University, Sweden

*Correspondence to: Claudia Mohr (claudia.mohr@aces.su.se)

**Keywords:** nanosecond laser ablation, femtosecond laser ablation, single particle mass spectrometry, atmospheric aerosols, aerosol mass spectrometry

**Abstract**

Size, composition, and mixing state of individual aerosol particles can be analysed in real time using single particle mass spectrometry (SPMS). In SPMS, laser ablation is the most widely used method for desorption and ionization of particle components, often realizing both in one single step. Excimer lasers are well suited for this task due to their relatively high power density ($10^7$ W cm$^{-2}$ - $10^{10}$ W cm$^{-2}$) in nanosecond (ns) pulses at ultraviolet (UV) wavelengths, and short triggering times. However, varying particle optical properties and matrix effects make a quantitative interpretation of this analytical approach challenging. In atmospheric SPMS applications, this influences both the mass fraction of an individual particle that gets ablated, as well as the resulting mass spectral fragmentation pattern of the ablated material. The present study explores the use of shorter (femtosecond, fs) laser pulses for atmospheric SPMS. Its objective is to assess whether the higher laser power density of the fs-laser leads to a more complete ionization of the entire particle and higher ion signal, and thus improvement of the quantitative abilities of SPMS. We systematically investigate the influence of power density and pulse duration on airborne particle (polystyrene latex, $SiO_2$, $NH_4NO_3$, NaCl, and custom-made core-shell particles) ablation and reproducibility of mass spectral signatures. We used a laser ablation aerosol time-of-flight single particle mass spectrometer (LAAPTOF, AeroMegt GmbH), originally equipped with an excimer laser (wavelength 193 nm, pulse width 8 ns, pulse energy 4 mJ), and coupled it to an fs-laser (Spectra Physics Solstice-100F ultrafast laser) with similar pulse energy, but longer wavelengths (266 nm with 100 fs and 0.2 mJ, 800 nm with 100 fs and 3.2 mJ, respectively). We successfully coupled the free firing fs-laser with the single particle mass spectrometer employing the fs-laser light scattered by the particle to trigger mass spectra acquisition. Generally, mass spectra exhibit an increase in ion intensities (factor 1 to 5) with increasing laser power density (~$10^8$ W cm$^{-2}$ to ~$10^{13}$ W cm$^{-2}$) from ns- to fs-laser. At the same time, fs-laser ablation produces spectra with larger ion fragments and ion clusters, as well as clusters with oxygen, which does not render spectra interpretation more simple compared to ns-laser ablation. The idea that the higher power density of the fs-laser leads to a more complete particle ablation and ionization could not be substantiated in this study. Quantification of ablated material remains difficult due to incomplete ionization of the particle. Furthermore, the fs-laser application still suffers from limitations in triggering it in a useful timeframe. Further studies are needed to test potential advantages of fs- over ns-laser ablation in SPMS.

## 1 Introduction

Atmospheric aerosols are known to have large impacts on climate change, air quality and human health, and these effects are strongly related to the chemical composition of individual aerosol particles (Fuzzi et al., 2015). Atmospheric aerosols are highly heterogeneous in composition, due to the vast number of natural and anthropogenic sources, as well as transformation and mixing processes during their residence time in the atmosphere (Kulkarni et al., 2011). Most analyses of aerosol chemical composition focus on the bulk, not least due to the very small mass and number of molecules present in an average atmospheric particle, making single particle studies challenging. However, mixing state and composition of individual particles are crucial information for the assessment of e.g. the particles' interactions with light or water vapour, and thus their contribution to climate change (Charles, 2012; IPCC, 2007; Jacobson, 2005; John, 2016; Laskin et al., 2018)

Single particle mass spectrometry (SPMS) is a powerful tool for the investigation of the size-resolved chemical composition of individual atmospheric aerosol particles in real time (Brands et al., 2011; Gaie-Levrel et al., 2012; Murphy, 2007; Murphy et al., 2006; Pratt et al., 2009; Pratt and Prather, 2012; Zelenyuk et al., 2010). Although a two-step approach separating laser ablation and laser ionization bears several advantages e.g. to identify specific molecules (Passig et al., 2017) currently still many instruments use a single step laser desorption and ionization. Single particle mass spectrometers currently in use by different groups worldwide have closely similar designs (Gaie-Levrel et al., 2012; Johnston, 2000; Murphy and Thomson, 1995; Zelenyuk et al., 2009). They usually consist of one or two scattering lasers to detect the particle size by particle time-of-flight, an ionization laser, commonly a nanosecond (ns) excimer laser for particle desorption and ionization in one single step, and the mass analyser (Murphy, 2007).

Quantitative analysis of single aerosol particles via laser ablation remains challenging, although several studies achieved advancements (Bhave et al., 2001; Fergenson et al., 2001; Gross et al., 2005; Healy et al., 2013), e.g. by detailed characterization of instrument sensitivity for individual chemical species, and by optimizing ionization laser parameters to reduce fragmentation. However, so far no single particle mass spectrometer is available for quantitative on-line analysis of particle mixtures. Single step laser desorption and ionization with Excimer lasers is highly nonlinear (Zelenyuk and Imre, 2005). Usually, particles are not completely ablated (Ge et al., 1996; Murphy, 2007), and the ablation process leads to irreproducible spectra. The dominant yield of species with low ionization potential further limits the quantitative ability of ns-laser SPMS (Reilly et al., 2000). The absorption of photons depends on the optical properties of the chemical components of the particle, with important implications for core-shell or multi-component particles (Cahill et al., 2015). Reported approaches for improvement of the quantitative abilities of SPMS include e.g. two step vaporization-ionization, where a $CO_2$ laser was used prior to excimer laser ionization for the evaporation of the particles (Cabalo et al., 2000; Morrical et al., 1998; Smith et al., 2002; Whiteaker and Prather, 2003; Woods et al., 2002), or the use of a high power density Nd:YAG laser of 5 ns pulse duration with 100 mJ pulse energy ($> 10^{10}$ W m$^{-2}$) (Lee et al., 2005; Mahadevan et al., 2002; Zhou et al., 2007). Lee and Mahadevan found that the kinetic energy of ions produced from ns-laser pulses is proportional to the particle size with a power law relationship, which influences ion detection efficiency in traditional time-of-flight mass spectrometer optics. The power density is much higher for femtosecond laser pulses compared to nanosecond laser pulses. The interaction of a high-intensity laser beam with a solid particle leads to the generation of a plasma that increases its energy content, its average charge state, and its charge density during the pulse

duration. Zhou et al. (2007) used a one-dimensional hydrodynamic model and experimental observations to explain ns-laser pulse interactions with particles, and observed that the laser energy absorption efficiency and thus ionization efficiency increases with shorter pulse width (from 10 ns to 10 ps). Molecular dynamic model simulations by Schoolcraft et al. (2001, 2000) explained the process of laser desorption and ionization of submicron particles for amorphous and crystalline particles with and without inclusions as a function of the nature

of the material.

Fs-lasers are widely used in the fields of micro machining and nanoparticle ablation (Chichkov et al., 1996; Gattass and Mazur, 2008; Malvezzi, 2014; Richard et al., 2013; Tsuji et al., 2003). Fs-laser ablation mechanisms include Coulomb explosion (soft ablation) and phase explosion followed by thermal ablation (strong ablation), depending on fs-laser intensities (Amoruso et al., 1999; Leitz et al., 2011; Roeterdink et al., 2003; Zhou et al.,

2007). Coulomb explosion and/or phase explosion happen in multiphoton ionization, depending on pulse energy. Because of the very short interaction time in the femtosecond laser ablation and ionization, it is considered to be almost instantaneous with a kinetic energy of the electrons which is sufficiently high for immediate escape from the target. Therefore, no space charge shielding of the sample should occur. Consequently, the target is left behind with a corresponding density of localised positive holes. Once a sufficient density of holes is achieved, the target

surface becomes electrostatically unstable, resulting in a Coulomb explosion of ions. The Coulomb explosion takes place in the initial phase and/or phase explosion occurs at a higher stage of multi photon ionization (Roeterdink et al., 2003). Various mechanisms of the fs-laser ablation (excitation, melting, ablation) were compared with the nanosecond laser ablation (Harilal et al., 2014; Malvezzi, 2014) at different time scales. Substantial atomization and strong cluster formation are the major effects due to the phase and/or Coulomb

explosion in the fs-laser ablation (Malvezzi, 2014; Roeterdink et al., 2003; Xu et al., 2000; Zaidi et al., 2010). The fs-laser ablation generates more atomic ions than in the nanosecond laser ablation due to rapid energy transfer, and also leads to formation of more ion clusters because of the explosions. A brief comparison between nanosecond and femtosecond laser ablation mechanisms for different timescales is given in the supplementary information (SI, Figure S1). Most of the processes discussed above are based on studies with solid substrates in

material science in which laser ablation is widely used. Only a few studies were done with single particles so far (Murphy and Thomson, 1995; Zhou et al., 2007). The ablation and ionization of airborne particles may be different from ablation of solid substrates e.g. with respect to energy dissipation within the substrate. However, the basic principle of laser-matter interaction should be similar in both the cases especially within the first nanoseconds.

Potential differences in the ablation mechanism and resulting single particle mass spectra as a function of pulse

duration were also a focus in our studies. Compared to ns-laser ablation, the pulse duration of ultrashort (pico- and femtosecond, fs) laser pulses is less than the typical picosecond time range of thermal diffusion effects. During ns-laser ablation thermal diffusion may reach deeper into the particles and the laser radiation may interact with the forming plume of ablated material. In contrast, for fs-laser ablation the plasma formed near the particle surface without deeper thermal diffusion generates a plume by Coulomb and/or phase explosion which is not interacting

with the laser light (cf. Fig. S1). The multi-photon ionization generates ions during the ablation phase which may undergo e.g. association reactions in the expanding plume. The Coulomb explosions can also lead to ions with high kinetic energy which can lead to broader peaks in the mass spectra (Henyk et al., 2000a, b). In the case of fs-laser ablation, the higher photon density may favour multi-photon ionization, which may lead to the formation of new species during Coulomb or phase explosion. However, the ion formation mechanism is not well understood.

Also for the widely used ns-lasers in SPMS, the ion formation mechanism is not completely understood (Murphy, 2007). Please note, the ablated particle components move up to ~5 μm during a 5 ns pulse or ~0.1 nm during a 100 fs pulse even, respectively, and in both cases remain well within the typical laser beam width. This estimate is based on an average ion speed of 1000 m s$^{-1}$ (Marine et al., 1992; Walsh and Deutsch, 1991).

Ultimately, the resulting mass spectral pattern in SPMS will depend on the energy transferred to the particle
via the ionization laser, which is related to pulse width, laser power, and wavelength. Shorter pulses exhibit e.g. higher power densities than longer pulses at the same laser energy. Higher power densities usually lead to higher ionization efficiencies, as e.g. in multiphoton interactions, where the total ion intensity is proportional to the power density of the ionization laser (Malvezzi, 2014). High-energetic, short-wavelength and short-pulse-duration ionization lasers may thus be a valid choice in single particle ionization.

In this study we explore the potential of high power density fs-laser pulses for improved quantitative abilities of SPMS. We compare mass spectra of atmospherically relevant airborne particles and dedicated test particles from ns- and fs-laser ablation using a commercially available laser ablation aerosol time-of-flight mass spectrometer (LAAPTOF, AeroMegt GmbH). The results presented here are an extension of the work by Zawadowicz et al. (2015), who coupled a Particle Analysis by Laser Mass Spectrometry (PALMS) with an fs-
laser (Spectra Physics Solstice-100F ultrafast laser) to investigate mass spectral patterns of NaCl, NH$_4$NO$_3$, and lead doped NH$_4$NO$_3$ particles. Overall, they observed similar mass spectra in both ns- and fs-laser ablation, but also showed that ions with high ionization energy such as Cl$^+$ are more easily generated by the fs-laser due to its higher power density. At the same time, their fs–PALMS exhibited lower sensitivity to lead than the PALMS employing the ns-laser.

To achieve a better understanding of fs-laser ablation in SPMS, we systematically obtained mass spectra of particles of different size, morphology (core-shell), and chemical composition for both ns- and fs-laser ionization. Please note that for this work the geometry of ablation/ionization laser beam particle interaction was not orthogonal as for the experiments described by Zawadowicz et al. (2015), but almost collinear as this was favoured by the design of the LAAPTOF. Furthermore, we varied the power density of the ionization lasers by varying the
laser energy, and laser wavelength in the case of fs-laser (800 and 266 nm), and by changing the focus positions of the lasers. We describe qualitative and quantitative differences of the mass spectra obtained, and discuss implications of our results for the quantitative abilities of the LAAPTOF, and SPMS in general.

## 2 Methodology
### 2.1 LAAPTOF

The LAAPTOF has been described already in several other publications (Gemayel et al., 2016; Marsden et al., 2016; Shen et al., 2017) and therefore we will focus here especially on those aspects related to its operation with the fs-laser. The LAAPTOF consists of four major modules: an aerodynamic lens focusing incoming particles into a narrow beam, the sizing section with two ultraviolet (UV) 405 nm continuous wave detection laser diodes
(DL1, DL2) set 11.3 cm apart from each other, the laser ablation/particle ionization region, and a bipolar time-of-flight mass spectrometer (Fig. 1). In the sizing region, the time delay between the detection of the scattered laser light of DL1 and DL2 by photomultiplier tubes (PMT, Thorn EMI, UK, type 9781R) is used to calculate the size of particles in the range of 200 - 2500 nm. The particle size detection methods and detection efficiencies for the

LAAPTOF have been described in several publications (Gemayel et al., 2016; Marsden et al., 2016; Shen et al., 2018). Shen et al. (2018) show a comparison of the performance of the instrument we used in this study with other and modified LAAPTOF instruments. The scattering signal from DL2 produces a 10 V transistor–transistor logic (TTL) electronic signal that triggers the 193 nm ns excimer laser (ArF, pulse width of 5 - 8 ns, adjustable pulse energy from 0 - 10 mJ, ATLEX-S, ATL Lasertechnik GmbH). The excimer laser shoots at the particle and at the

same time triggers the data acquisition system with a 5 V TTL electronic signal. We did vary the laser focus to the left/right and up/down and determined the diameter of the particle beam to 1 - 2 mm, depending on particle type. The ns-laser beam is slightly defocused at the position (F1) increasing the particle-laser interaction area, and the defocused beam diameter is 99±31 μm where it encounters the aerosol particle (F1, Fig. 1). The focus position of the excimer laser is at 20 cm from the lens, and ionization happens 3 - 4 cm after the focus position, for F2 and

F1, respectively. This is the distance from focus point to the centre of the ion extraction region from where the ions are extracted into the mass analyser. The movable lens can be used to shift the focus position from F1 to F2 where the defocused beam diameter is 81±7 μm resulting in higher power densities acting on the particles. Please note that the position of the ionization region is quite well defined in this case, close to the centre of the ion extraction zone, due to the scattering signal of the second detection laser whereas for the experiments with the fs-

laser we had to apply a different procedure to define this (see section 2.2 and 3.1). Variation of the focus position allows to vary the power density by a factor of ~1.5 for otherwise similar conditions, for F1 and F2, respectively. The laser beam produces both positive and negative ions, which are deflected into the corresponding time-of-flight regions of the bipolar mass spectrometer. Typically, each particle hit by the excimer laser generates a positive and negative mass spectrum.


## 2.2 Fs-laser coupled LAAPTOF

    The fs-laser (Spectra Physics Solstice-100F ultrafast laser) we coupled to the LAAPTOF is a Ti:Sapphire source, emitting pulses of 800 nm radiation with 1 kHz. Pulse duration is ~100 fs. The laser beam profile is close to

Gaussian with a beam quality factor $M^2 < 1.3$ ($M^2 = 1$ for an ideal Gaussian beam). The maximum pulse energy is ~3.5 mJ. We also used a third harmonic generation module (Spectra Physics, TP-THG-F) to generate 266 nm pulses of 0.2 mJ and 100 fs duration. However, the resulting one order of magnitude smaller pulse energy of the laser compared to the default 800 nm wavelength led to a reduced light scattering signal, corresponding ineffective triggering of mass spectra recording, and thus reduced particle detection and lower ion signal. We therefore focus

our analysis in this manuscript on fs-laser spectra at 800 nm wavelength. In this work, the peak power density (calculated as the power per beam area at focal point) is varied by changing the pulse energies. Excimer (ns-) and fs-laser beam parameters as well as pulse energies and corresponding peak power densities at focus points F1 and F2 (see below) are listed in Tables 1 and 2.

    For the coupling of the fs-laser with the LAAPTOF, a few technical changes were necessary. Fig. 1 also

includes a schematic of the modified LAAPTOF. Since it was not possible to trigger the fs-laser, it was running in free firing mode with a frequency of 1 kHz. To only record spectra from when a particle was hit by the fs-laser, the scattered light from the fs-laser interaction with the particle detected by the second set of PMT was used to trigger the data acquisition. This and the fact that scattered light of the high-power fs-laser could be detected by the PMT (yielding false particle size information) meant that both detection lasers were futile, and thus switched

off. This led to absence of size information. To avoid loss of spectral signal due to the delay of ~10 μs between

triggering and the start of data acquisition, data acquisition was run in pre-trigger mode (Fig. 2). To define the ionization region for this case also close to the centre of the ion extraction region a procedure selecting those mass spectra with more than 90% of the maximum total ion intensities was applied (cf. section 3.1).

A movable focusing lens set-up was used for multiple focusing positions between F1 and F2 further towards inlet, to better understand the effect of power density on mass spectral patterns (insert in Fig. 1). The laser beam diameters are calculated for all three wavelengths and for two different focus positions (cf. Table S4). For the wavelength of 800 nm the laser beam diameters are 487±77 µm and 246±36 µm at the positions F1 and F2, respectively. The focal positions were varied to study the effect of power density on the mass spectra. The power densities at F2 are ~3.5 times higher than at F1.

**2.3 Particle types and experiments**

For comparison of mass spectral patterns and signal intensities of fs- and ns-laser ablation in the LAAPTOF, we chose the following particle samples: monodisperse polystyrene latex (PSL) particles, core-shell particles with a gold core and shells of silver (Ag), silica ($SiO_2$), and poly(allylamine hydrochloride) (PAH), salts (NaCl, $NH_4NO_3$), and spherical silica ($SiO_2$) particles. The sample details are tabulated in Table S3. All samples were dissolved or diluted in nano pure water (18 MΩ), nebulized (Topas ATM 221, Topas AG), and then dried with silica driers (Topas AG). The dried aerosol particles were size-selected with a differential mobility analyser (DMA 3080, TSI) and sent to the LAAPTOF.

All samples were measured with different laser power densities corresponding to different energies for both ns- and fs-lasers (Table 2). Per sample and laser type and setting, we tried to record spectra from at least 500 particles (see Table S1 and S2 in the SI). Empty spectra were excluded in the data analysis.

**3 Results and Discussion**

**3.1 Qualitative differences between ns- and fs-laser spectra**

In the following, the qualitative differences between ns- and fs-laser positive and negative mass spectra are compared for typical pulse energies (4 mJ in the ns-laser and 3.2 mJ in the fs-laser, respectively). The mass spectra that are discussed in this section were selected to be representative for each particle type in the following manner. From the typically 200-600 useful single particle mass spectra measured for each particle type only those 30-40% (60-240 spectra) with at least 90% of the maximum total ion intensities were selected to ensure optimal hit of the particles by the ablation and ionization laser. These remaining spectra were classified using the fuzzy c-mean algorithm available in the LAAPTOF data analysis software. This resulted in typically two classes of mass spectra per particle type. For each class of mass spectra we manually selected 10 spectra representing all main characteristics and applied an additional mass axis calibration for each spectrum. These 10 spectra showed correlation coefficients of r = 0.7-0.9. An example demonstrating the reproducibility and representativeness of this selection process is given in the SI (cf. Figs S2-S5). Analysis of mass spectra for both polarities from this work resulted in mass resolutions at full width half maxima for masses 16, 24 and 48 of 458, 530, and 593, respectively. At this resolution we can distinguish peak differences on a single mass unit basis. Please note that

240 most difficulties in peak assignment don't originate from mass resolution, but from the jitter of the mass axis from spectrum to spectrum or particle to particle.

### 3.1.1 PSL particles

245 A comparison of typical mass spectra of individual PSL particles with a geometric diameter ($d_p$) of 500 nm for ns- and fs-LAAPTOF is shown in Fig. 3. The positive ns-laser spectrum consists of series of carbon clusters $C_n^+$ (n=1–7) and hydrogenated carbon clusters $C_nH_m^+$ (n = 1–7, m = 1–3) with different intensities (± 10 %) for different single particles. The insert in Fig. 3a shows the clusters with 5 carbon atoms. All other carbon clusters have similar numbers of hydrogen atoms. The negative ns-spectrum exhibits generally lower signal intensity and 250 consists only of three major $C_n^-$ (n = 1 – 3) and $C_nH_m^-$ (m = 1–2) peaks.

With the fs-laser (800 nm, similar results for 266 nm, not shown), a total of ~ 500 particle spectra were recorded. From these 500 particles, 54 % are of type 1 (Fig. 3c-d), and 42 % of type 2 (Fig. 3e-f), respectively. Type 1 consists of series of positive and negative $C_n^+$, $C_nH_m^+$ (n=1–8, m=1–2), $C_n^-$ and $C_nH_m^-$ (n=2–8, m=1–2) ions. The positive spectrum also exhibits signal at m/z 18 and 16, which we assign to $H_2O^+$ and $O^+$, potentially 255 from residual water. The insert in Fig. 3c shows the clusters with 5 carbon atoms ($C_5H_m^+$, m = 0–2). The ion patterns are similar for positive ion clusters with different carbon atom numbers. Type 2 spectra consist of a longer series of carbon-containing clusters $C_n^+$ and $C_nH_m^+$ (n=1–15, m=1–2) in positive mode. Signal intensity is lower in negative mode, with shorter carbon cluster series $C_n^-$ (n=2–3). The type 1 spectra show more negative ion clusters than the type 2 spectra, whereas type 2 shows more positive ion clusters. One explanation for this 260 observation could be that the type 2 spectra are generated from particles that are ionized closer to the positive ion extraction region, whereas the type 1 spectra may arise from particles ionized closer to the negative ion extraction region or in the middle of the ion extraction region of the mass spectrometer. Since the particle beam at the ionization region has a width of 1-2 mm and the laser beam a width between 487±77µm (F1) and 246±36 µm (F2) it is possible that some particle are ionized closer to either one of the electrodes leading to these two types of mass 265 spectra.

In both laser ablation methods we observe formation of carbon clusters and hydrogenated carbon cluster ions from PSL particles. For fs-laser ablation, larger carbon clusters (> 7 carbon atoms) with (in positive mode) fewer hydrogen atoms (< 3 hydrogen atoms) are observed. In both laser ablation methods we observe formation of carbon clusters and hydrogenated carbon cluster ions from PSL particles. For fs-laser ablation, larger carbon 270 clusters (> 7 carbon atoms) with (in positive mode) fewer hydrogen atoms (< 3 hydrogen atoms) are observed. Such larger clusters in the fs-laser spectra can potentially form during the Coulomb or phase explosion of the fs-laser ablation process but some studies claim that also reactions of the primary ion species with the source plume can generate the larger clusters (Zaidi et al., 2015; Zaidi et al., 2010). For both laser pulse durations, the number of larger clusters increased with increasing laser pulse energy for the PSL spectra as has also been reported for 275 ns-laser pulses by Weiss et al., (1997).

### 3.1.2 NaCl particles

Bipolar mass spectra of NaCl particles of an electrical mobility diameter ($d_m$) of 400 nm for ns- and fs-laser are shown in Fig. 4. The positive ns-laser spectrum consists of atomic sodium and molecular ions ($Na^+$, $Na_2^+$), and $Na_2Cl^+$ and $Na_3Cl_2^+$ ions. The negative ion spectrum consists of chlorine ions ($Cl^-$, $Cl_2^-$), and sodium chloride cluster ions ($NaCl_2^-$, $Na_3Cl_2^-$). The bipolar spectrum is representative of 55% of a total of 600 spectra. The remaining spectra are empty or contain no features of NaCl.

From the fs-LAAPTOF measurements (800 nm, similar results for 266 nm, not shown) ~ 80 single particle spectra were considered for analysis. This low hit rate for the fs-laser compared to the ns-laser may be related to both, the particle shape widening the particle beam and the reduced absorption at 800 nm compared to 193 nm. A representative single particle bipolar spectrum is shown in Fig. 4c-d. The positive spectrum consists of $Na^+$, $Na_2^+$, $Cl^+$, $Na_2Cl^+$, $NaCl_2^+$, $Na_3Cl_2^+$, $NaO_2^+$ ions, and $H^+$, $O_2^+$ ions most likely from residual water. The negative spectrum consists of $Na^-$, $Na_2^-$, $Cl^-$, $Cl_2^-$, $Na_2Cl^-$, $Na_2Cl_3^-$, and $Na_4Cl_4^-$ ions. Around 20 % of particles exhibited ~ 10 % less intense signal of $Na^{+/-}$ and $Na_2^{+/-}$, while the signal of the other ions was similar across the spectra.

The qualitative difference between ns- and fs-LAAPTOF spectra is the presence of $Cl^+$ and $Na_2^+$ ions in positive, and $Na_2^-$ and $Cl_2^-$ ions in negative fs-spectra, which are not commonly observed in ns-spectra. In addition, $NaO_2^+$ and $NaO_2^{2+}$ ions (O likely from residual water) are present in the positive spectrum of the fs-LAAPTOF. Zawadowicz et al (2015) reported ions with similar combinations of Na and Cl in their fs-PALMS single particle spectra, but no NaO clusters. The dissociation energy of the NaCl molecule is 4.26 eV, thus smaller than the ionization energy of $Na^+$ (5.13 eV) and $Cl^+$ (12.96 eV). The electron affinity of $Na^-$ and $Cl^-$ is 0.52 eV and 2.35 eV, respectively (Sansonetti and Martin, 2005). In the case of ns-laser ablation with lower power density, NaCl can easily dissociate and form $Na^+$ and $Cl^-$ ions. For $Cl^+$ ion formation, twice as much energy (12.96 eV) is required, available during the fs-laser ablation process. Ionization energies are even higher for molecular Na and Cl, as well as for doubly charged Na and Cl ions. We also observed more cluster ions in the fs-laser spectra ($Na_2Cl^+$, $NaCl_2^+$, $Na_2Cl^-$, $Na_2Cl_3^-$ and $Na_4Cl_4^-$) compared to the ns-laser spectra, which again indicates a more complex ionization mechanism during fs-laser ablation. Several studies on fs-laser ablation of NaCl have observed the formation of cluster ions at higher power densities due to Coulomb or phase explosion, depending on excitation energy (Hada et al., 2014; Henyk et al., 2000a, b; Reif et al., 2004).

### 3.1.3 NH₄NO₃ particles

The positive and negative mass spectra of $NH_4NO_3$ particles with $d_m = 400$ nm for ns- and fs-LAAPTOF are shown in Fig. 5. The ns-laser spectrum is representative of 500 particles. The positive spectrum consists of $NH_2^+/O^+$, $OH^+$, $NH_4^+$, $NO_2^{2+}$, $NO^+$, $NO_2^+$, and $NO_3^+$ ion signatures, while the negative spectrum contains $N^{2-}$, $NH_2^-/O^-$, $OH^-$, $NH_4^-$, $NO_2^{2-}$, $NO^-$, $O_2^-$, $NO_2^-$, $NO_3^-$, and $HNO_3NO_2^-$ ions.

The fs-LAAPTOF (800 nm) bipolar spectrum represents only 10 % of a total of 500 spectra. The majority of the particles was poorly hit and/or ionized. The positive spectrum contains $H^+$, $N_2^+$, $(NH_4)_2^+$, and $(NH_4)_3^+$ ions in addition to the ions observed in the ns-LAAPTOF positive spectrum, however there is no $NO_3^+$ signature. There is much lower (two orders of magnitude) signal intensity in the negative spectrum compared to the positive spectrum. Only ~1 % of negative fs-laser spectra exhibit significant signal (panel d), with peaks from $N^{2-}$, $N_2^-$, $NO^-$, $O_2^-$, and $NO_2^-$ ions, with albeit lower sensitivity.

The ns-LAAPTOF bipolar spectrum is comparable to the ammonium nitrate single particle spectrum obtained by the PALMS (Zawadowicz et al., 2015). $NH_4NO_3$ predominantly leads to positive ions ($NH_2^+/O^+$, $OH^+$, $NH_4^+$,

$NO_2^{2+}$, $NO^+$, and $NO_2^+$ are the most intense peaks). Clear signal was observed with the ns-LAAPTOF for the **$HNO_3NO_2^-$** ion, in accordance with observations with an on-line laser desorption/ionization (LDI) mass spectrometer with excimer laser wavelength 193 nm (Neubauer et al., 1998), but in contradiction to observations with the PALMS (Zawadowicz et al., 2015) or the real-time single-particle mass spectrometer (RSMS) with excimer laser wavelength 193 nm (Reinard and Johnston, 2008). The fs-LAAPTOF positive spectrum consists of

$N_xO_y^+$ (x = 0-2, y = 0-2), $(NH_4)_x^+$, where x = 1-3, $H^+$, and $N_2^+$ ions. The ionization energies of $(NH_4)_x^+$ (for x > 1) cluster ions are very high (Dunlap and Doyle, 1996), which may be the reason for the absence of these ions in the ns-spectra.

### 3.1.4 SiO$_2$ particles


Mass spectra of individual SiO$_2$ particles of $d_p$ = 1000 nm measured by ns- and fs-LAAPTOF are shown in Fig. 6. A bipolar spectrum of SiO$_2$ particles, representative of about 80% of 860 spectra from ns-LAAPTOF obtained with 4 mJ pulse energy is presented in panels a and b. The positive spectrum consists of $Si^+$ and $SiO^+$ ions, the negative spectrum contains $O^-$, $SiO_2^-$, and $SiO_3^-$ ions. The other 20 % of spectra have three more $Si_xO_y^-$ clusters (x

= 2-3, y = 4-5).

Recent studies with a LAAPTOF by Marsden et al. (2016) of silicate (SiO$_4$) rich ambient dust particles featured similar mass spectral peaks, namely from $Si^+$, $SiO^+$, and $O^-$, $SiO^-$, $SiO_2^-$ ions. Single particle characterization studies of SiO$_2$ rich particles by Cahill et al. (2015) with an Aerosol Time-of-Flight Mass Spectrometer (ATOFMS) using a 266 nm Nd:YAG ionization laser also produced $O^-$, $SiO_2^-$, and $SiO_3^-$ ions. Both studies

presented similar spectra to the ones shown in Fig. 6a – b. Another ambient single particle study with a PALMS powered with an excimer laser (193 nm, ~4 mJ pulse energy) by Gallavardin et al. (2008) showed $Si_xO_y$ negative ion clusters (x = 1-2, y = 1-4), similar to the peaks in 20 % of spectra from our LAAPTOF (not shown here). Another experimental study of silica clusters with a 308 nm XeCl excimer laser coupled time-of-flight mass spectrometer observed clusters of silica $(SiO_2)_n^-$ units (n = 1-6) (Xu et al., 2000).

The fs-LAAPTOF bipolar spectra for silica are shown in Fig. 6c – d, representative of ~ 80 % of 530 particles. The positive spectrum consists of $O^+$, $Si^+$, $O_2^+$, $Si_xO_y^+$ (x = 1-3, y = 2x+1), and $Si_xO_y^+$ (x = 1-2, y= x+1) ions, the negative spectrum of $O^-$, $Si^-$, $O_2^-$, $Si_xO_y^-$ (x = 1-6, y = 2x+1), and $Si_xO_y^-$ (x = 1-2, y = x+1) ions. To our knowledge there are no studies on fs-laser ablation of individual silica particles, we thus cannot compare our spectra with other single particle spectra. Kato et al. (2007) investigated fs- and ns-laser ablation of silica substrates, and

silicon-rich solutions. They observed positive silicon clusters, $Si_x^+$ (x = 1-6) and claimed that ns-laser ablation leads to more fragmentation, whereas fs-laser ablation leads to better atomization with elemental ionization. With fs-laser ablation of silicon substrate, clusters of $Si^+_n$ (n = up to 10) have been observed (Bulgakov et al., 2004).

The major difference between the positive ns-LAAPTOF and fs-LAAPTOF spectra is the existence of elemental oxygen ($O^+$, $O_2^+$), and silica clusters ($SiO_2^+$, $Si_2O_3^+$, $Si_2O_5^+$, and $Si_3O_7^+$) in the fs-laser spectra. In the

negative fs-laser spectra, $O^-$ and $Si^-$ elemental ions as well as $Si_xO_y^-$ (x = 1-6, y = 2x+1) and $Si_xO_y^-$ (x=1-2, y= x+1) cluster ions are observed. Only $O^-$, $SiO_2^-$, and $SiO_3^-$ ions are also common in ns-laser negative spectra. Another major difference is the high intensity of signal in the fs-LAAPTOF spectra. Overall, fs-laser ablation

yields more elemental information (positive silicon and positive oxygen), but also leads to higher-order clusters than ns-laser ablation. The increasing abundance of larger clusters with increasing laser pulse energy is shown in Fig. S6 for $SiO_2$ particles.

Since the $SiO_2$ is bonded covalently in nature, high energies are required to break it. Consequently, we observe both cations and anions of $SiO_2$ constituents in the fs-spectra. The 266 nm UV fs-LAAPTOF spectra of $SiO_2$ particles contain similar spectral features (cf. Fig. S10), but less intense signal than the 800 nm fs-laser spectra.

### 3.1.5 Gold-silver core-shell particles

A bipolar single particle mass spectrum of gold-silver core-shell particles from ns-LAAPTOF is shown in panels a and b of Fig. 7. This bipolar spectrum represents 27 % of a total of 850 particles or spectra. The peaks below 100 Th are fragments from the surfactant Cetyl-trimethylammonium bromide (CTAB), $C_{19}H_{42}BrN$, which was used to stabilize the gold-silver particles in the suspension. The positive spectrum consists of signal from the two silver isotopes $Ag^{(107,109)+}$, of $Au^+$, and $AuO^+$. The negative spectrum contains signal from $Ag^{(107,109)-}$, $Au^-$, silver oxide isotopes $AgO_2^{(139,141)-}$, and also silver dioxide $(AgO_2)^-$. About 43 % of the 850 spectra contain only signal from elemental silver and silver oxides, and no gold peaks. The remaining 30 % of single particle spectra consist only of signal from CTAB.

The respective fs-LAAPTOF bipolar spectrum is shown in Fig. 7c-d, which is representative of 19 % of ~ 1500 particles. The positive spectrum consists of signal from the surfactant CTAB, elemental silver isotopes $Ag^{(107,109)+}$, and silver oxide isotopes $Ag^{(139,141)+}$. The negative spectrum is almost empty, with very small peaks of silver and silver oxide, and none of the fs-LAAPTOF spectra contains gold signal. 8 % of the negative fs-LAAPTOF spectra have clear signal from the surfactant and very small silver peaks.

The gold-silver particles contain a 300 nm gold core (nearly 67 % of total the weight percentage with a mass of 2.18 pg), and a 150 nm thick shell of silver (about 33 % of the total mass percentage with a mass of 1.08 pg). The ns-LAAPTOF mass spectral signatures, however, feature higher signal for silver than for gold. More signal from both core and shell is observed when there is less surfactant signal in the ns-spectra. The major difference between ns-LAAPTOF and fs-LAAPTOF spectra is the absence of gold peaks in the fs- spectra. At the wavelength of the excimer ns-laser (193 nm), the reflectance of both gold and silver are nearly 35 % (Kah et al., 2015). At the wavelength of the fs-laser (800 nm), the reflectance is more than 95% for both gold and silver. The high reflectance of gold and silver in the IR likely contributes to the reduced ablation of the core. Although the reflectance of these particles is much lower at 266 nm these fs-laser pulses were also not capable to generate a significant signal from the core for the reduced pulse energy of 0.2 mJ.

The existence of gold and silver oxides in both ns- and fs-laser spectra may be explained by oxygen containing coatings on the particles e.g. of water or surfactants. However, we can't fully exclude interactions of the ablated plume with background ions or residual water like in the study by Neubauer et al. (1998).

### 3.1.6 Gold-Poly(allylamine hydrochlorid) core-shell particles

The second type of of core-shell particles we tested for mass spectral comparison between ns- and fs-laser is made of a gold core ($d_p$ = 300 nm) and an organic polymer (poly (allylamine hydrochloride), PAH) shell (coating thickness = 50 nm). Ns- and fs-LAAPTOF spectra are shown in Fig. 8.

The ns-LAAPTOF bipolar spectrum (panels a and b) is representative of 44 % of totally 450 particles. It features signal from the organic PAH shell (elemental carbon $C_n^{+/-}$, and hydrocarbons $C_nH^{+/-}$), as well as elemental gold ($Au^{+/-}$) ions. The remaining 56 % of spectra mainly consist of signal from the PAH shell, and exhibit no or almost no signal from the gold core.

Two types of fs-LAAPTOF spectra are shown in panels c-f. The spectra are representative of 10 and 15 % of the total number of spectra (1000), respectively. Spectra that do not contain gold or PAH signal are excluded, as are spectra that contain only background signal peaks ($Ar^+$, $CO_2^+$). Fig. 8c-d is representative of a spectrum that contains distinctive signal from the gold core, and $C_n^+$ ions as well as $C_nH_n^+$ cluster ions. The second type (panels e-f) does not exhibit any signal from the gold core, and also less signal from the carbon clusters. The spectra with both organic shell and gold core signature are most likely produced from particles hit very close to the centre of the laser beam. The spectra without gold signature are mot likely produced from particles interacting only with part of the laser beam. Please note that the particle beam has a diameter ranging between 1-2 mm while the laser beam diameter ranges between 246 and 487 µm (cf. Table S4).

The comparison of ns- and fs-LAAPTOF spectra of gold-PAH core-shell particles reveals more signal from carbon clusters, which are similar to the peaks from PSL particles, in the fs-spectra. Signal from the gold core was only observed in one of a total of 1000 particles, again due to the low absorption of gold in the IR (Manca et al., 2007; Pereira et al., 2015). The spectra of gold-PAH particles from the 266 nm fs-LAAPTOF (not shown) also did not exhibit signal from the gold core, and only low signal intensity from the PAH shell.

The third type of core-shell particles, Au-SiO$_2$, produced mass spectra and no signal from the gold core at both fs-laser wavelengths (Fig. S2). Signal from the gold core was again observed in the ns-laser spectra. They contained Si, SiO, $SiO_2$ positive ions, as well as hydrocarbon ions from the surfactant, and negative hydrocarbon, elemental carbon, and oxygen signal from the $SiO_2$ shell.

**3.1.7 fs-laser ablation with a wavelength of 266 nm**

The mass spectra obtained for fs-laser pulses of 266 nm wavelength and 0.2 mJ energy/pulse show very similar features for all the samples measured as obtained for the other fs-laser wavelength of 800 nm. Please note that about 80% of the spectra collected for 266 nm were empty due to reduced light scattering signal and corresponding ineffective triggering of mass spectra recording. Furthermore, the mass spectra containing information have a 3 to 5 times lower average intensity for all particle types compared to those obtained for fs-laser pulses of 800 nm, with a similar energy of 0.3 mJ per pulse. A discussion on single particle mass spectra for 266 nm (fs-laser) in comparison with those obtained using 193 nm (ns-laser) and 800 nm (fs-laser) is presented in the SI (Figs. S7 - S13).

**3.2 Signal intensity as a function of laser power density and particle size**
**3.2.1 Signal intensity variation with laser power density**

We investigated the relationship between ion signal intensity and laser power density for all particle samples. The increase in power density from the lowest (0.8 mJ) to the highest (8 mJ) excimer laser pulse energy corresponds to one order of magnitude ($2.06*10^8 – 2.06*10^9$ W cm$^{-2}$). For the fs-laser, the energy was varied from 0.3 mJ to 3.5 mJ, corresponding to more than one order of magnitude difference in power density. Due to the large spectrum-to-spectrum variance, average signal intensities per power density settings were calculated for 20 mass spectra representative of the typical spectra discussed in section 3.1 (cf. Fig. S2).

The mass spectra generated with the fs-laser have on average a factor of 5 higher total ion intensity compared to those generated with the ns-laser. Average signal intensity versus power density for both ns- and fs-lasers is shown in Fig. 9. All samples exhibit an increase in average signal intensity (by a factor of up to 5, depending on particle type) with increasing ns- or fs-laser (800 nm) power. The exception is NaCl, which seems to be more efficiently ionized (compare section 3.1.2) at the higher power densities of the fs-laser compared to the ns-laser, albeit with a saturation effect. Based on our limited data and the available literature one can only speculate about potential reasons. The observed slight saturation effect of signal intensity at higher power densities for both lasers and most particle types may be due the Coulomb repulsion among the ions during multiphoton ionization, observed as well by L'Huillier et al. (1987). Furthermore, the penetration of the plasma into the particles with increasing power density may be limited e.g. due to absorption of part of the additional power by the plasma near the surface.

Fig. S14 shows the same data as Figure 1, but separated for focus positions F1 and F2. Note that for the fs-laser, due its free firing mode, the ionization position and corresponding power density is highly uncertain and represents a best estimate. Consequently, we cannot rule out an overlap between possible power densities corresponding to F1 and F2, respectively.

### 3.2.2 Ion signal intensity variation with particle size

To explore the quantitative abilities of the fs- and ns-laser we also investigated the average ion signal intensity variation as a function of laser power density with respect to particle size (subplots a and b in Fig. 10), using PSL particles of 500 and 1000 nm diameter. Similar subplots (Fig. 10c – d) are shown for focus position F1 with lower power density. The average signal intensity for the 1000 nm size particles as a function of the excimer laser power density is 2-4 times higher compared to the signal intensity for 500 nm particles for both focus positions. The femtosecond laser produced only 1.5 -2 times larger average ion signals for 1000 nm particles compared to 500 nm particles. However, this difference between ns- and fs-laser ion intensities for these different particle sizes is within the uncertainties and also has to be verified for different types of particles. The mass ratio of the two particle sizes is 8, hence much larger than the relative differences in the total ion intensities. The ratio of the surface area of the 1000 nm PSL and 500 nm PSL particles is 4 which is comparable to the maximum intensity difference observed. This could be an indication that the ionization scales with the particle surface area. The increase in ion signal thus does not scale linearly with the difference in mass of the two particles sizes and of the total material potentially to be ablated. Similar effects were observed for RbNO$_3$ and (NH$_4$)$_2$SO$_4$ particles (Reents et al., 1994). This demonstrates the quantitative limitations of both ns- and fs-laser ablation.

### 4 Conclusions

We coupled the commercially available single particle mass spectrometer LAAPTOF, originally equipped with an ns excimer laser, with a free firing fs-laser to investigate mass spectral patterns and signal intensity for a variety of atmospherically relevant aerosol particles and dedicated test particles. We successfully employed the fs-laser light scattered by the particles to trigger mass spectra acquisition with a certain pre-trigger. Particle types sampled (and their diameter) include PSL particles (500 nm, 1000 nm), NaCl (400 nm), $NH_4NO_3$ (400 nm), Silica (1000 nm), and gold-silver, gold-SiO$_2$, and gold-PAH core-shell particles (600 nm, 400 nm, and 400 nm, respectively).

Overall, mass spectral signatures for ns-and fs-laser ablation and different particle types are fairly similar. Generally, ns-laser spectra for the same particle type exhibit higher reproducibility of the spectral pattern than fs-laser spectra with the LAAPTOF in counter propagating geometry. This is most likely because the ns-laser can be triggered, which leads to some limitation of the physical extent of the potential interaction region of particle and laser beam, which is not the case for the free firing fs-laser. Please note that between 30 to 40% of the spectra obtained using the fs-laser have the same spectral features, which demonstrates the reproducibility within a single type of measurement and which is a good basis to compare results for different measurement conditions. Qualitative differences between fs- and ns-laser spectra vary depending on particle type. Larger clusters ($C_nH_m$ clusters with $n > 7$ for PSL particles, or higher order $SiO_2$ clusters) were e.g. observed in the higher power density fs-laser spectra. Such larger clusters in the fs-laser spectra can potentially form during the Coulomb or phase explosion of the fs-laser ablation process. Some studies claim that also reactions of the primary ion species with the source plume may generate the larger clusters (Zaidi et al., 2010). However, these complex processes of fs-laser ionization are beyond the scope of this paper but require further studies. We find that these large clusters in fs-laser spectra do not necessarily improve the quantitative abilities of SPMS. For NaCl particles, only in fs-laser spectra high ionization energy species like $Cl^+$ or $N^+$ were detected. Fs-laser ablation also led to formation of oxides, for e.g. core-shell particles, silica particles, and silver oxides for the gold-silver core-shell particles.

Apart from differences in ionization process, also laser wavelength and particle optical properties play an important role in SPMS, especially important for core-shell particles or inhomogeneously mixed particles. However, for fs-laser ablation it seems that the rapid plasma formation on the surface e.g. of the core-shell particles prevents deeper impact and hence ablation and ionization of core material at least for shell thicknesses of 150 nm. The mass spectra available from the fs-laser with 266 nm and an energy of 0.2 mJ have shown very similar spectra as for the fs-laser operating with 800 nm and 0.3 mJ. Despite the relative small number of usable spectra for 266 nm we consider it very likely that high power densities and hence multi-photon ionization taking place for both wavelengths lead to the formation of similar ions which points to similar ion formation mechanisms. However, a more detailed discussion of possible ion formation mechanisms is not possible based on the data available.

Generally, fs-laser generated mass spectra show a factor of 1 to 5 higher total ion intensities compared to those from the ns-laser. Variation in power density does not have a large impact on mass spectral patterns for both laser types, but influences ions signal intensity. The average ion signal intensity is increased by a factor of 2 - 5 for an increase in laser power density by at least one order of magnitude for both laser modes. Ion signal intensity also shows an (albeit non-linear) dependency on particle size, as tested for PSL particles with diameters of 500 and 1000 nm. This non-linearity would warrant further investigation for these instruments to improve their quantitative abilities. The smaller impact of particle size on ion signal intensity for the fs-laser compared to the ns-laser system observed here indicates that fs-lasers might not be the most effective way to improve SPMS quantification. High

energetic (~100 mJ) nanosecond lasers may potentially be a better choice than high power density fs-lasers due to the operational ease and cost.

The idea, that the higher power density that can be achieved with fs-laser pulses leads to a more complete particle ablation and ionization, could not be substantiated in this study. However, the cluster formation nature of fs-laser ablation rewards more studies with aerosol particles to understand and correlate the results for potential improvements in quantification and mixing state analysis. Further tests including e.g. two step ionization or delayed extraction are needed to investigate potential advantages of fs- over ns-laser ablation in atmospheric SPMS.

*Data availability*

Data will be provided upon request by the authors.

*Author contributions*

R.R. set up the experiment, did the measurements, analysed the data and wrote the manuscript. A.A. operated the femtosecond laser system and did the optical coupling. X.S. helped with the setup. H.S., T.L., and C.M. developed the scientific approach, and supported the experimental procedures and data analysis. R.R., H.S., and C.M. developed the manuscript. All co-authors participated in scientific discussions on the interpretation of the results.

*Acknowledgements*

The Authors would like to thank Daniel J. Cziczo and his team for their work which was base for our project. The Authors would like to thank Denis Duft for scientific discussions, Divya Kumar for laser beam profile calculations and Georg Scheurig, Steffen Vogt, and Frank Schwarz for their technical support. This work was funded by the Ministerium für Wissenschaft, Forschung und Kunst in Baden-Württemberg, Germany in the program Research Seed Capital. AA is grateful to the German Research Foundation for support (DFG, AB 604/1-1 and AB 604/1-2).

*Competing interests*

The authors declare that they have no conflicts of interest.

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

**Table 1: Parameters of the ns- and fs-lasers: λ = wavelength, E= energy per pulse, $\tau$ = pulse duration, beam diameters before focussing lens and at the two interaction positions F1 and F2.**

| Parameter | λ (nm) | E (mJ) | t (ns) | f (Hz) | Laser beam diameter Incident | Laser beam diameter $(\mu m)$ | |
|---|---|---|---|---|---|---|---|
| | | | | | | At F1 | At F2 |
| Excimer –ns Laser (ArF) | 193 | 0.4 - 8 | ~5-8 | Max 300 | 24 mm | $99\pm31$ | $81\pm7$ |
| fs- Laser Ti:sapphire | 800 | ~0.3-3.5 | ~0.10 | 1000 | 7 mm | $487\pm77$ | $246\pm36$ |
| fs- Laser Ti:sapphire | 266 | ~0.2 | ~0.10 | 1000 | 7 mm | $270\pm32$ | $182\pm32$ |

**Table 2: Excimer ns-laser and fs-laser pulse energies and corresponding power densities at positions F1 and F2. The corresponding beam diameters are given in Table S4.**

| Laser Type | Excimer ns-UV Laser (193nm) | | | fs-UV (266nm) | fs-IR (800nm) | | | |
|---|---|---|---|---|---|---|---|---|
| Energy (mJ) | 0.8 | 4.0 | 8.0 | 0.2 | 0.3 | 1.7 | 3.2 | 3.5 |
| Peak power density at F1 (W cm$^{-2}$) | $2.06\times 10^9$ | $1.03\times 10^{10}$ | $2.06\times 10^{10}$ | $3.49\times 10^{12}$ | $1.61\times 10^{12}$ | $9.12\times 10^{12}$ | $1.72\times 10^{13}$ | $1.88\times 10^{13}$ |
| Peak power density at F2 (W cm$^{-2}$) | $3.11\times 10^9$ | $1.55\times 10^{10}$ | $3.11\times 10^{10}$ | $7.69\times 10^{12}$ | $6.32\times 10^{12}$ | $3.58\times 10^{13}$ | $6.74\times 10^{13}$ | $7.37\times 10^{13}$ |





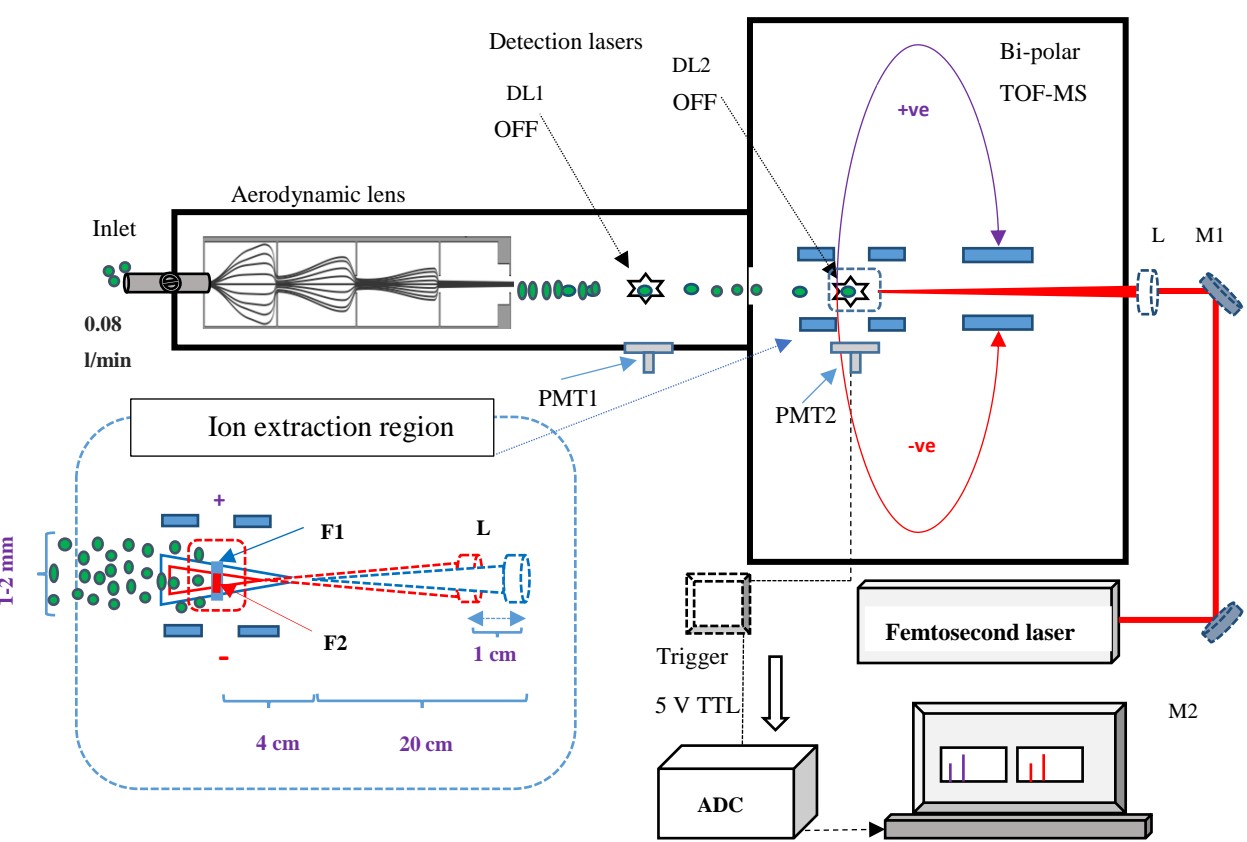

**Figure 1: Schematic diagram of the LAAPTOF coupled with the fs-laser. The detection lasers DL1 and DL2 are turned off, as well as the first set of PMT (PMT1). PMT2 collects the scattered light from the interaction of the fs-laser with the particle. The PMT2 signal is connected to the trigger box and produces a trigger with a 5 V TTL signal for the ADC data acquisition. M1, M2 are mirrors, and the aerosol particles are shown in the green dots. The corresponding spectra of each ablated particle are recorded and stored in the computer. The inset picture shows the centre of ion extraction region (dashed red square) and the variation in laser beam diameters for the two different lens or focus positions (F1 blue & F2 red). The laser focus is 3-4 cm before the ion extraction region. The counter propagating particle beam (green dots) has a diameter of 1-2 mm depending on particle type. Moving the lens 1 cm towards the ion extraction region (F2) reduces the beam diameter and increases the power density at the centre of the extraction region. (Picture is not to the exact scale).**

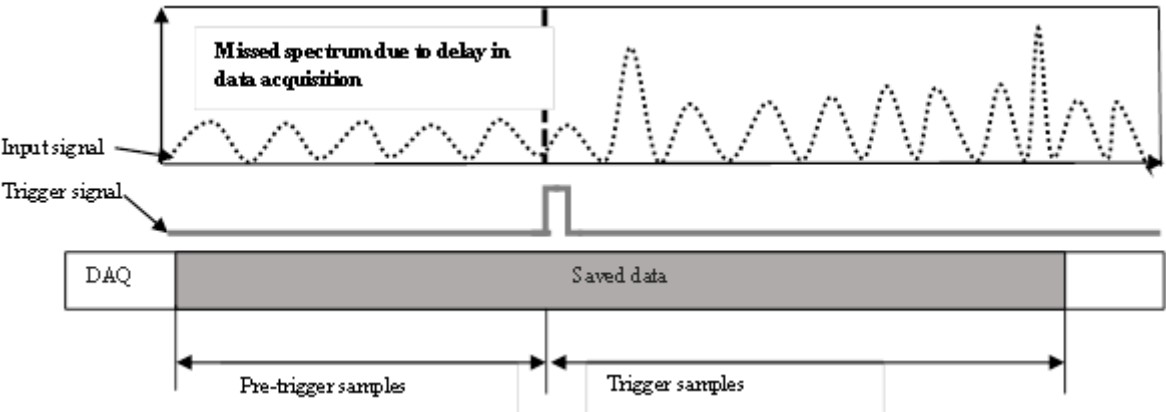

**Figure 2: Pre-trigger sampling mechanism of the data acquisition system. The data (input signal, the dotted wavy line) arriving before the trigger event (pre-trigger samples) are saved in the temporary memory of the data acquisition card and then combined with the trigger samples.**

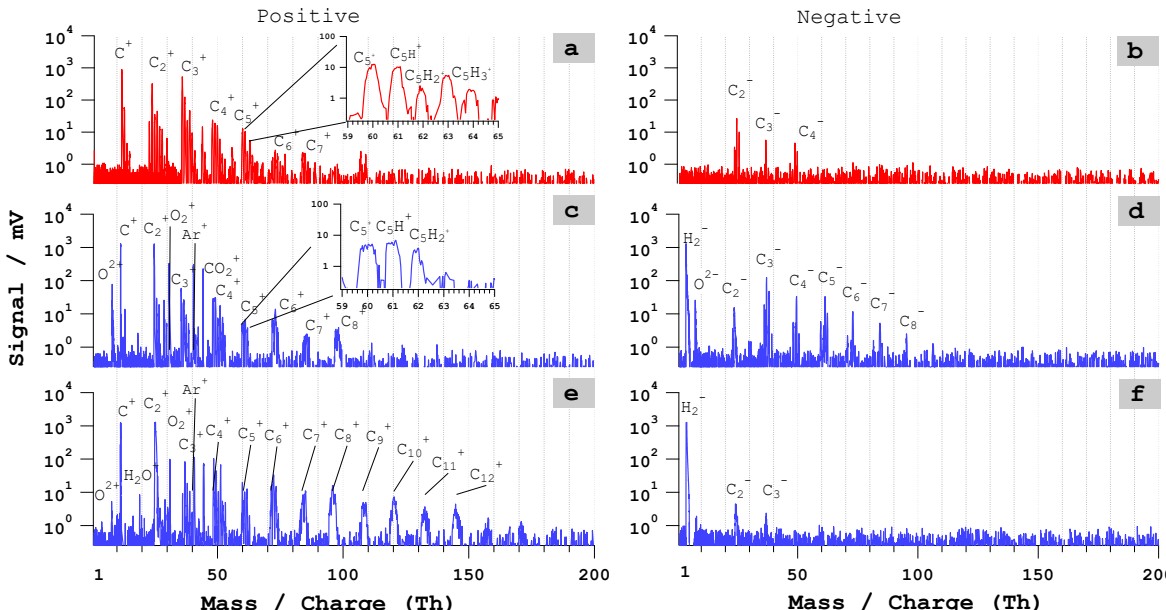

**Figure 3: Bipolar mass spectra of single PSL particles of $d_p$= 500 nm. a – b: Typical ns-laser spectra; c – d: fs-laser ($\lambda$ = 800 nm) spectra for 54 % of particles (type 1); e –f: fs-laser ($\lambda$ = 800 nm) spectra for 42 % of particles (type 2). The pulse energy is 4 mJ for the ns-laser and 3.2 mJ for the fs-laser.**

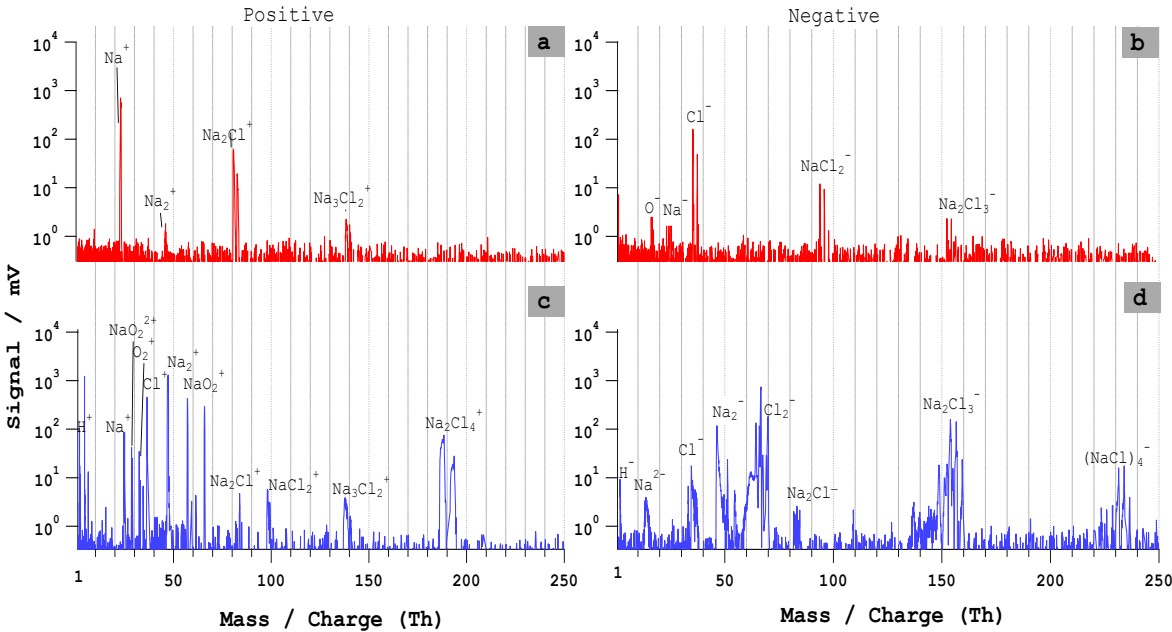

**Figure 4: Bipolar mass spectra of NaCl single particles of $d_m$ = 400 nm. a – b: Typical ns-laser spectra; c – d: fs-laser (λ = 800 nm) spectra. The pulse energy is 4 mJ for the ns-laser and 3.2 mJ for the fs-laser.**

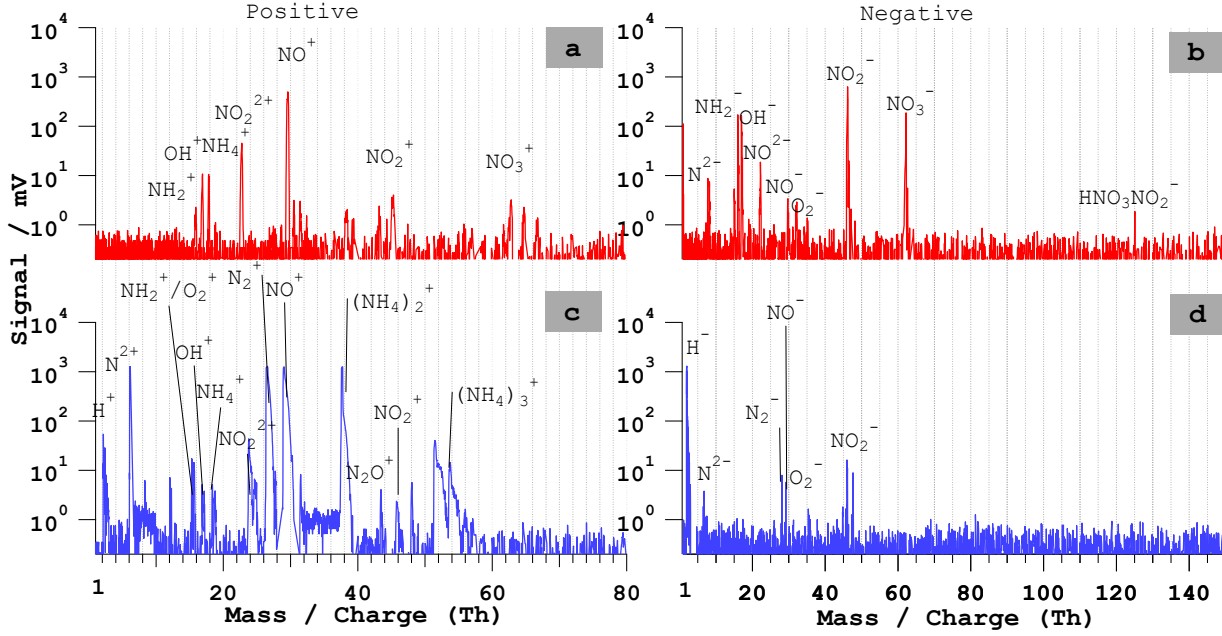

**Figure 5: Bipolar mass spectra of NH₄NO₃ particles of $d_p$ = 400 nm. a – b: Typical ns-LAAPTOF spectra; c – d: fs-LAAPTOF (λ = 800 nm) spectra. The pulse energy is 4 mJ for the ns-laser and 3.2 mJ for the fs-laser.**

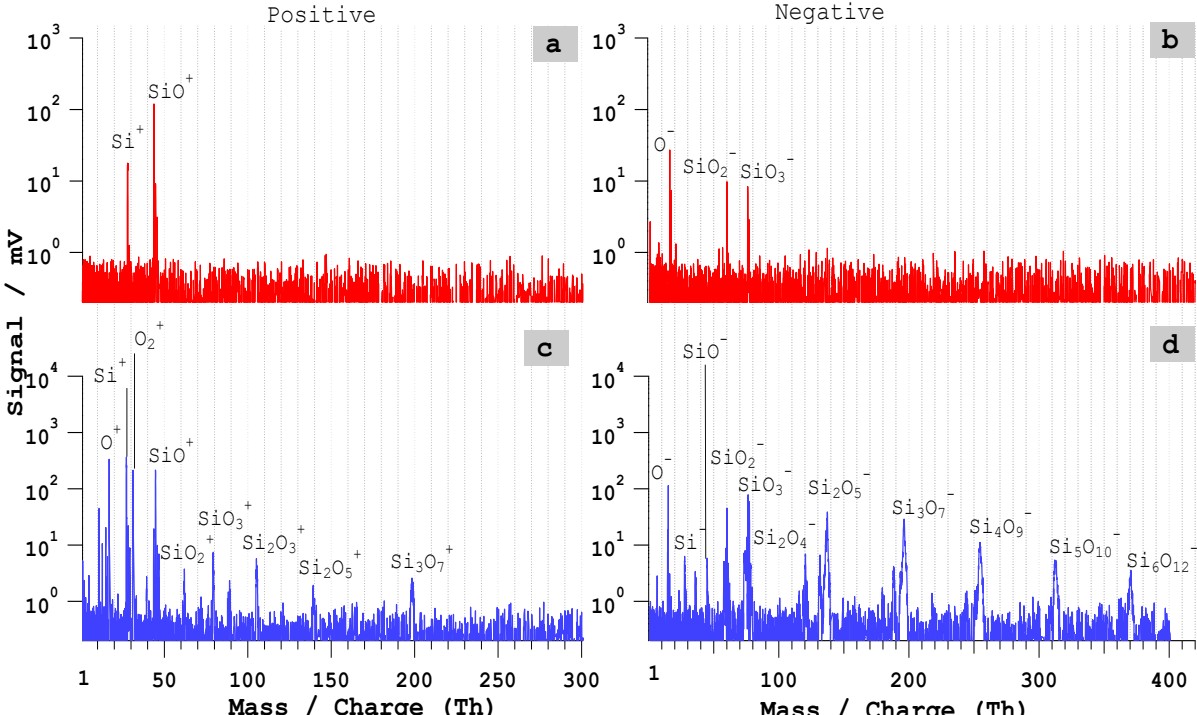

**Figure 6: Bipolar mass spectra of SiO₂ single particles of d_p= 1100 nm. a – b: Typical ns-laser spectra; c – d: fs-LAAPTOF (λ = 800 nm) spectra. The pulse energy is 4 mJ for the ns-laser and 3.2 mJ for the fs-laser.**

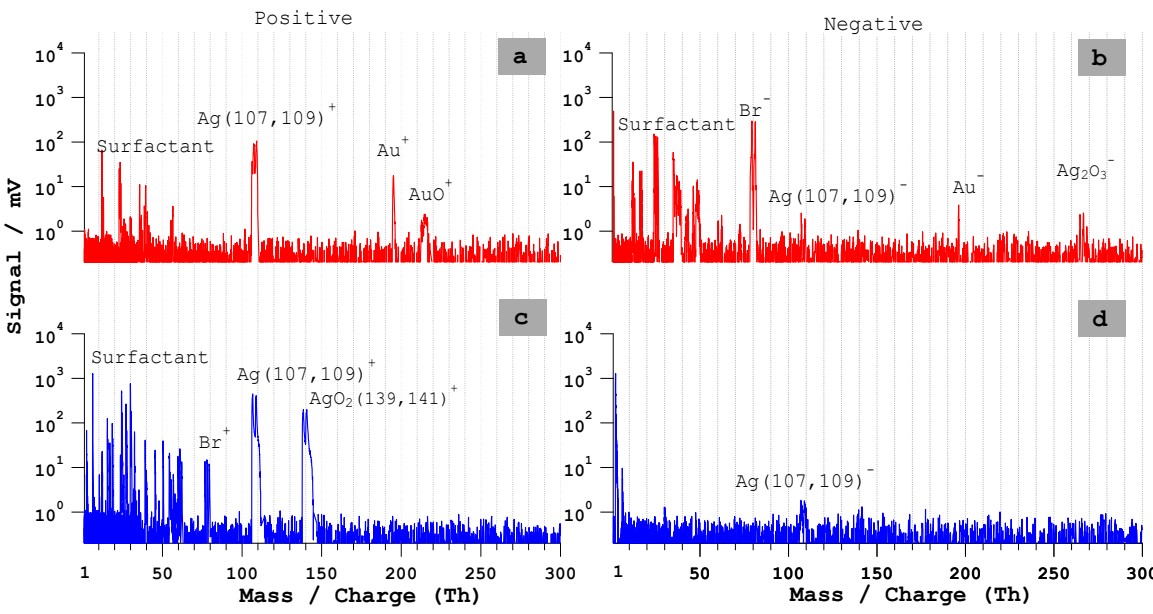

**Figure 7: Bipolar mass spectra of gold-silver core-shell particles of dp= 600 nm. a – b: Typical ns-LAAPTOF spectra; c – d: fs-LAAPTOF (λ = 800 nm) spectra. The pulse energy is 4 mJ for the ns-laser and 3.2 mJ for the fs-laser.**

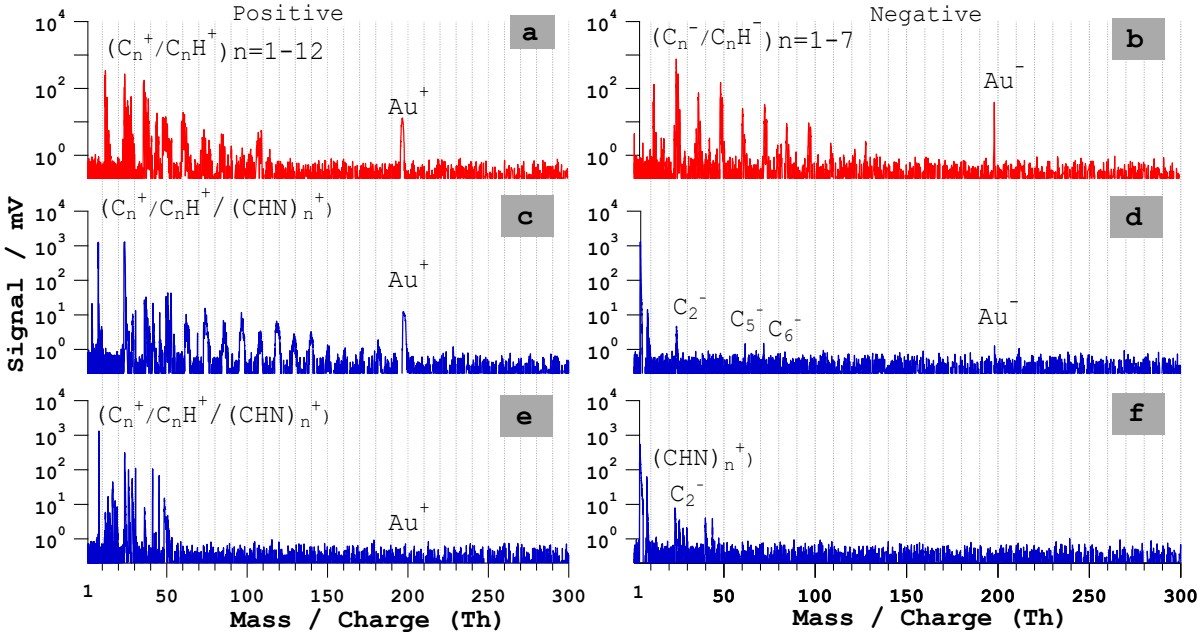

**Figure 8: Bipolar mass spectra of Au-PAH core-shell particles. a – b: Typical ns-LAAPTOF spectra; c – d: fs-LAAPTOF (λ = 800 nm) spectrum that contains both gold and PAH signal, e-f: fs-LAAPTOF (λ=800 nm) spectrum without gold signal. The pulse energy is 4 mJ for the ns-laser and 3.2 mJ for the fs-laser.**

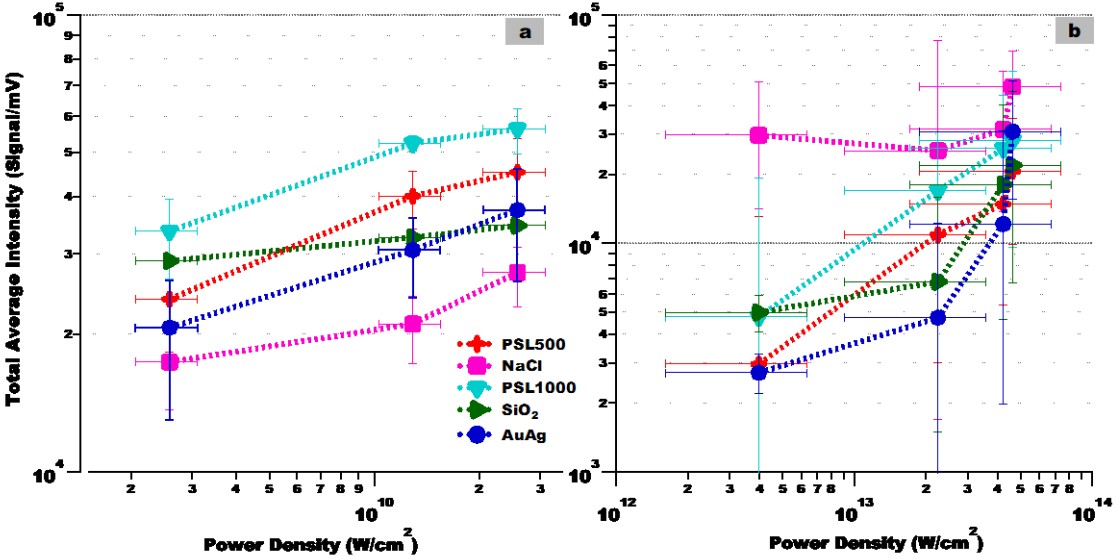

**Figure 9: Average ion intensity versus laser power density for different particle types. (a) Excimer laser, (b) fs-laser. Error bars of the average ion signal intensity correspond to ±1 standard deviation, error bars of the power density to the value of power density of the laser beam at the corresponding possible maximum and minimum position of particle-laser interaction.**

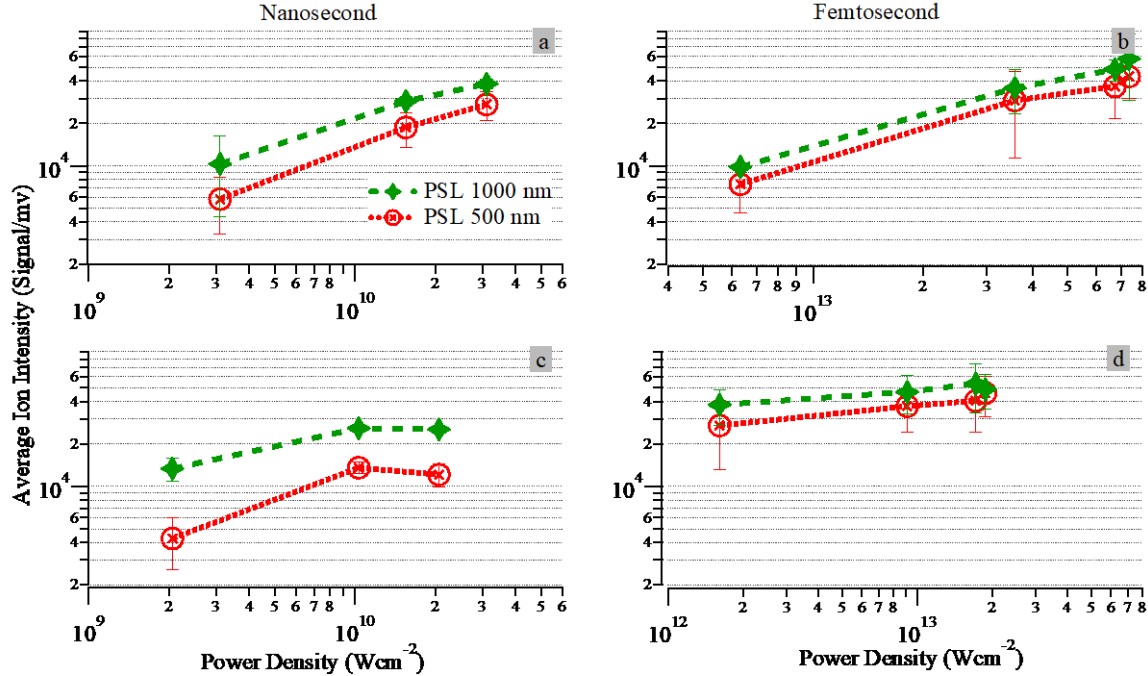

**Figure 10: Variation of average total ion intensity with respect to the size of PSL particles. The total ion intensities were averaged for 10 representative spectra of each particle type. (Particle diameters: 500 nm, red circles, and 1000 nm, green triangles). (a, b) at focus position F2 and (c, d) at focus position F1.**

830