# Peer review of "Exploring femtosecond laser ablation in single particle aerosol mass spectrometry"

_Atmospheric Measurement Techniques, 2017_

## Referee Comment (RC1) · Anonymous Referee #1 · 3 Dec 2017

**General Comments**

The authors present a comparison of the mass spectra obtained from the laser desorption ionisation of atmospherically relevant particle types using ns and fs laser systems in a LAAPTOF single particle mass spectrometer. Although a similar study of has been presented previously by Zawadowizc (2015), this manuscript dealt with a different instrument geometry, and therefore the study presented here has the potential to contribute to the understanding of laser-particle interactions in single-particle mass spectrometry.

However, the authors fail to emphasise the key difference in geometry between the LAAPTOF and the PALMS instrument in the previous study i.e the counter propagate vs orthogonal arrangement of the excimer laser with respect to the particle beam axis. This difference in geometry is likely to influence how the ablation process proceeds and will also influence the interpretation of the spectral characteristics for the reasons stated in the specific comments below.

In addition, the manuscript has some major deficiencies which need to be addressed. With the exception of the ion intensity comparison of the two systems, the key conclusions offered in the abstract in this manuscript are not supported in the data or discussion. Namely:

- There is no supporting data for the conclusions regarding the reproducibility of spectral patterns.
- The methods used for quantifying the fraction of spectra types is not provided.
- The conclusion that 'fs-laser ablation produce spectra with larger ion fragments and ion clusters, as well as clusters with oxygen, which does not render spectra interpretation more simple compared to ns-laser ablation' is only supported by a single hand-picked spectra of each particle type.
- There is no evidence or discussion of the claim in the abstract that quantification remains difficult due to the incomplete ablation of the particle.

Evidence of a rigorous statistical comparison of the mass spectral patterns is required.

**Specific comments**

P1, L23. Reproducibility is not specifically discussed in this manuscript.

P2, L64. Do you mean a solid particle or fixed target/substrate? Please clarify.

P3, L96. This is only applicable to positive $Cl^+$ ions. $Cl^-$ ions are readily observed in negative in mode due to high electron affinity.

P3, L199 and P4, L1. Were the beam diameter and focal length/position measured or calculated?

P4, L3. Do you mean the excimer focal position? How do you know ionisation happens 3-4cm after the focal position with a counter propagate geometry? Can you comment on the depth of field of the focusing? Is it more likely that ionisation takes place before the focal position? Does the position of ionisation depend on the absorbing properties and ionisation threshold of the material with a counter propagate geometry?

P4, L129. What is the diameter of the focal point of the fs system?

P4, L141. Can the authors comment on the stability/reproducibility of the focal length, the depth of field and the effect on power density? How is the power density different at F1 than F2? Surely the densities are the same and the focal position has just shifted upstream in the particle beam. Why do the authors vary the focal length? The objectives and conclusions of this operation are not stated in the manuscript.

P5. L171 and throughout the manuscript. How were the mass spectra assigned to a spectra type?

P7, L243. Silicate particles predominantly come from mineral dust not sea spray. $SiO_4$ is an anion in crystalline orthosilicates and is not an appropriate description of silicate composition.

 P7, L256. The work cited used **Silicon** substrate with organic and inorganic solutions and the claims regarding the fragmentation/ clusters were relating the compounds in solution not the silicon. Please be careful not to confuse clusters with fragments.

P10, L361. It should be pointed out that the conclusions are with respect to SPMS with a counter propagate geometry.

P10, L361. Do you mean reproducibility of average ion intensity or spectral pattern?

P10, L369. $Cl^-$ was observed in negative ion spectra. See comment above.

**Technical Corrections**

P2, L92. GmbH not Gmbh.

P10, L372. Sentence structure/word order.

Figure1 Inset. It is not clear what the position label, defocus label and arrow are referring to.

Figure 2. It is not clear what the wavy line represents.

Figures 3-9. Please state the estimated power density or laser setting at which these spectra were acquired.

Figures 9 and 10. Please state how many particles were averaged and what the error bars represent. The error bars are difficult to see and the quality of the graphics are generally poor.

---

## Referee Comment (RC2) · Anonymous Referee #2 · 10 Feb 2018

**General comments**

This paper deals with the single particle mass spectrometry (SPMS) of atmospherically relevant aerosol particles (or laboratory surrogates) in the one-step (ablation/ionization) laser approach, comparing the performances of ns (193 nm) and fs (800 nm and 266 nm) laser irradiation. As a preliminary comment, although this technique can be useful for classifying atmospheric particles based on their (fragmentation) mass spectral fingerprints, it does not provide detailed molecular information on the actual chemical composition of the particles. The two-step (separate ablation and ionization lasers) approach is much more effective on this (see, e.g., Zimmermann's group studies, Anal. Chem. 2017, 89, 6341). The introduction should better acknowledge this, i.e. mention the actual chemical analysis capabilities, not only the improvement of quantitative

abilities in SPMS.

The paper presents an important amount of experimental data. However, it is written mostly at a descriptive and comparative level, with no deeper insight into the actual mechanisms involved in the ablation and ionization processes. Purely speculative assertions (e.g. "... which again indicates a more complex ionization mechanism during fs-laser ablation", rows 212-213, or rows 179-182, 334-336 etc.), cannot contribute much to advancing our knowledge on this technique. Moreover, as the authors acknowledge themselves, this paper is an extension of a previous one (Zawadowicz et al., Anal. Chem., 2015), the difference being an "in-line" laser irradiation, compared to the orthogonal one used in the previous paper. The choice of this new configuration is not justified by the authors. It introduces an important experimental uncertainty on the actual ablation/ionization position in the ion source of the bipolar mass spectrometer (evaluated by the authors at 2-4 cm from the focus – how was this calculated?), which further generates a lack of precision in the discussion (see, e.g., rows 179-182). Additionally, this results into significant uncertainty on the laser irradiance actually experienced by the particles at the ablation/ionization spot. The reported laser irradiances, calculated in the focal plane, are therefore mostly useless for comparison with experiments performed by other groups in different geometries. The conclusion of the paper is that the use of a fs laser presents rather limited interest, when compared to much common (and cheaper) ns sources. This conclusion can be a bit rushed. Although not evident here, the fs approach can still have an interesting application in depth profiling of phase-separated or mixed aerosol particles, but for this a precise and reproducible alignment of the laser beam with respect to the particle must be achieved, which is clearly not the path followed in this study.

It is very surprising that the two wavelengths (800 nm and 266 nm) used for fs ablation/ionization returned absolutely similar results. On one hand, the multi-photon ionization (MPI) invoked by the authors is very different at the two wavelengths (three times more photons needed in IR to reach the same ionization energy), this should result in orders of magnitude difference in the MPI yield, which could not be compensated by the 20-fold higher IR energy/pulse. On the other hand, the optical properties of the studied particles (although extensively mentioned in the Introduction) are not properly used in the text to account for the experimental observations. Most of the discussion (e.g. high reflectance of Au at 800 nm) is based on single-photon interaction assumptions, while at the high intensities reached in fs regime everything is so multi-photonic (i.e. non-linear). Moreover, the optical properties at 266 nm are completely ignored.

Specific comments

1. Rows 68-69: "The energy per unit volume is greater for femtosecond laser pulses compared to nanosecond laser pulses" – why? The energy/pulse is comparable for ns and fs (800 nm). Is the focusing different? Can you clearly specify the beam diameter at the focus (or better, at the interaction) spot for all three beams used? For each experiment: please indicate clearly the energy per pulse used.

2. Rows 81-83: "In the case of fs-laser ablation, the higher photon density may favour multi-photon ionization, which may lead to the formation of new species from the ablated plume in subsequent Coulomb or phase explosion" – the formulation is not clear. Please state clearly what processes are taking place in the condensed phase (particle) and which ones in the gas phase (plume).

3. Rows 120-122: "beam diameter is ∼300 $\mu$m" – how was this value obtained? Calculated/measured? If calculated, were the lens aberrations taken into account? If the same lens was used for 193 nm and 800 nm, how the focal length and all subsequent calculations change between these two wavelengths? What is the error bar on the beam diameter? Error bars should be indicated also on the irradiance values all over the manuscript (including Figures). How was the 2-4 cm position after the focus determined? Does this translates into 1-3 cm for F1 focusing? How these values change between the three wavelengths, considering the change in the focal length? F1 and F2 mentioned here are not indicated in Figure 1. Please give the beam diameter limits in

the (2-4 (1-3) cm ?) laser-particle interaction region for all wavelengths, this would be much more useful than diameter at the focus. Indicate also the irradiance limits with the error bars related to calculations and measurements.

4. Please report mass resolution for both polarities. From Figures, this seems to be around 100. In these conditions, how certain can be the assignment of some mass peaks, e.g. m/z 16, 18?

5. Rows 179-182: the explanation for observation of type 1 vs type 2 spectra is not convincing. Can the authors provide a more developed explanation, based on experimental evidence? Generally speaking, a more thorough discussion on type 1 vs type 2 spectra is needed (see also comment 12 below), as this can have practical implications on particle classification in "real world" (field) experiments.

6. Rows 183-188: formation of larger carbon clusters for fs-ablation: "This may be due to the higher power density of the fs-laser, and reactions of the primary ion species with the source plume forming larger clusters as secondary products" – what is the experimental evidence for the in-plume growth of these clusters? How their intensity changes with the increase in laser irradiance? Please show the data (at least in Supplementary Information), they must be available from studies performed in section 3.2. In the Conclusion sections, the in-plume reactions are not mentioned, but only formation at the ablation stage (rows 366-367). Please put in agreement the conclusions with the main text assertions.

7. Section 3.1.2: the optical properties of NaCl particles are quite well-known and should be used to explain the low efficiency in generating mass spectra in fs AND ns regimes.

8. Rows 211-213: please clarify what you mean. Are these species generated in the ablation process, or by subsequent interactions in the plume? What is the role of the ionization here?

9. Section 3.1.3: an explanation should be advanced for the very low efficiency in generating mass spectra in both positive (10%) and negative (1%) polarities with the (800nm?) fs laser, with respect to the much higher (100%?) efficiency achieved with the ns one.

10. Section 3.1.4: less intense signal at 266 nm compared to 800 nm – please try to relate this to the optical properties of the SiO2 particles.

11. Rows 290-292: "The high reflectance of gold in the IR likely leads to reduced ablation of the core" – beside the fact that the absorption processes in the fs regime must be highly multi-photonic, this conclusion is questionable, as similar spectra are observed for 266 nm fs irradiation, or at this wavelength the reflectance of gold and silver is much reduced (∼30%). How can the authors interpret this? The same explanation is given rows 311-312, although the same similarity is observed between 800 nm and 266 nm irradiation.

12. Rows 305-309: an explanation for the existence of two types of spectra must be advanced.

13. Row 329: is this average factor relevant, considering the huge variability in efficiently generating usable mass spectra?

14. Rows 334-336: "... saturation effect ... may be due to coulombic repulsion ..." – please develop. Why this effect would occur only for NaCl particles? How this saturation correlates with the low efficiency (16%) in generating non-empty mass spectra with the fs laser? How is this saturation effect related to the optical properties of the NaCl particles (vs the others)?

15. Row 345: factor 7 claimed is not evident from Fig. 10, please check.

16. Row 348: Factor 8 in volume is not "much larger" than factor 7 in ion intensity (if confirmed).

17. Rows 350-351: "This demonstrates the quantitative limitations of both ns- and fs-

laser ablation". However, can the authors infer something about the fraction of particle mass which is vaporized from the measured data in Figure 10?

Technical corrections

18. Rows 80-81: ablated particles cannot move 5 $\mu$m during 5 ns and 0.1 $\mu$m during 100 fs, please check

19. Rows 301-303: 44% + 66% = 110%

20. Rows 387-388: please check English

21. Tables 1 and 3 can go to Supplementary Information

22. Table 2 is useless in this form, everyone can apply the proportionality on the energy/pulse. Give instead proper beam diameters in the interaction zone for each wavelength (see above)

23. Fig. 10 caption: inversion red-green

---

## Author Comment (AC1) · 23 Mar 2018

**Reply to Reviewer 1**

*We gratefully thank the reviewer for the careful manuscript reading, and the constructive comments which were helpful to improve the quality of our manuscript. Our point-to-point replies are given below in italics and in blue following the original comments:*

**General Comments**

The authors present a comparison of the mass spectra obtained from the laser desorption ionization of atmospherically relevant particle types using ns and fs laser systems in a LAAPTOF single particle mass spectrometer. Although a similar study of has been presented previously by Zawadowizc (2015), this manuscript dealt with a different instrument geometry, and therefore the study presented here has the potential to contribute to the understanding of laser-particle interactions in single-particle mass spectrometry.

However, the authors fail to emphasize the key difference in geometry between the LAAPTOF and the PALMS instrument in the previous study i.e. the counter propagate vs orthogonal arrangement of the excimer laser with respect to the particle beam axis. This difference in geometry is likely to influence how the ablation process proceeds and will also influence the interpretation of the spectral characteristics for the reasons stated in the specific comments below.

*It is correct that in this study we used a different geometry for laser ablation compared to the previous study by Zawadowizc et al., 2015, but furthermore it was our aim to study different particle types (e.g. pure organic, salts and core shell) for different power densities to get a better understanding of the light particle interaction. The principle difference is indeed that in our set up the particles interact with different intensities of the laser beam. However, we did obtain only reasonable signal intensity if the particles were hit close to the focus of the laser beam, and selected those mass spectra that had not less than 90% of the maximum total ion intensity. Due to this procedure we ensured that the laser intensity acting on the particle was reproducible and close to the maximum possible. Mass spectra of particles that were hit only partially, or only partially ionized, e.g. if they were not close enough to the focus, were omitted. However, we cannot determine exactly how close to the focus position the particles were ionized. We did vary the laser focus to the left/right and up/down and determined the diameter of the particle beam to 1-2 mm, depending on particle type, while the diameter of the laser focus was calculated for all wavelengths to range between 81-487 μm as listed in Table S4. We added the following sentence to the introduction:*

*"Please note that for this work the geometry of ablation/ionization laser beam particle interaction was not orthogonal as for the experiments described by Zawadowicz et al. (2015), but almost collinear as this was favoured by the design of the LAAPTOF."*

In addition, the manuscript has some major deficiencies which need to be addressed. With the exception of the ion intensity comparison of the two systems, the key conclusions offered in the abstract in this manuscript are not supported in the data or discussion. Namely:

- There is no supporting data for the conclusions regarding the reproducibility of spectral patterns.

- The methods used for quantifying the fraction of spectra types is not provided.

- The conclusion that 'fs-laser ablation produce spectra with larger ion fragments and ion clusters, as well as clusters with oxygen, which does not render spectra interpretation more simple compared to ns-laser ablation' is only supported by a single hand-picked spectra of each particle type.

- There is no evidence or discussion of the claim in the abstract that quantification remains difficult due to the incomplete ablation of the particle. Evidence of a rigorous statistical comparison of the mass spectral patterns is required.

*As discussed above we selected for our analysis mass spectra with at least 90% of the maximum total ion intensities obtained for each particle type, which led to a selection of 30-40 % of the total of 200-600 individual spectra obtained for each particle type, hence 60-240 spectra. From these spectra we selected two typical average spectra classes by a fuzzy c-mean algorithm available within the LAAPTOF data analysis software. For each class of mass spectra we selected 10 representative spectra showing all main characteristics and applied an additional mass axis calibration for each spectrum.*

*An example demonstrating the reproducibility and representativeness of the selected spectra has been added in the supplementary section for one particle type (Figure S3- S6).*

*We added the following text to section 3.1 of the manuscript:*
*"The mass spectra that are discussed in this section were selected to be representative for each particle type in the following manner. From the typically 200-600 useful single particle mass spectra measured for each particle type only those 30-40% (60-240 spectra) with at least 90% of the maximum total ion intensities were selected to ensure optimal hit of the particles by the ablation and ionization laser. These remaining spectra were classified using the fuzzy c-mean algorithm available in the LAAPTOF data analysis software. This resulted in typically two classes of mass spectra per particle type. For each class of mass spectra we manually selected 10 spectra representing all main characteristics and applied an additional mass axis calibration for each spectrum. These 10 spectra showed correlation coefficients of r = 0.7-0.9. An example demonstrating the reproducibility and representativeness of this selection process is given in the supplementary section (cf. Figures S3-S6)."*

*Consequently, we consider our conclusions that fs-laser ablation produces spectra with larger ion fragments and ion clusters, as well as clusters with oxygen compared to ns-laser ablation to be based on representative mass spectra.*

Mainly in section 3.2.2 and in the conclusions we discuss the potential quantification abilities of the fs laser ablation single particle mass spectrometer. Our study shows that there is only relatively small difference in total ion intensities between ns- and fs-laser ablation and ionization (section 3.2.1). Furthermore, the experiment with the core shell particles clearly show no increase in signal from the core for the fs-laser (3.1.5-6). We therefore believe our conclusion that the quantification of ablated material remains difficult due to incomplete ionization of the particle to be justified.

**Specific comments**
P1, L23. Reproducibility is not specifically discussed in this manuscript.
*An example demonstrating the reproducibility and representativeness of the selected spectra has been added in the supplementary section for one particle type (cf. Figures S3-S5). One example shows two raw mass spectra of PSL 500 nm particles for two different measurements of the same energy per pulse and the same focus position resulting in the same characteristic spectra for both measurements. Furthermore, 20 mass spectra for 20 different particles are compared showing the same spectral characteristics.*

*Hence we consider it justified to speak of "reproducibility of mass spectral signatures" in the abstract. Furthermore, we have added the following lines to the conclusions.*

*"Please note that between 30 to 40% of the spectra obtained using the fs-laser have the same spectral features, which demonstrates the reproducibility within a single type of measurement and which is a good basis to compare results for different measurement conditions."*

P2, L64. Do you mean a solid particle or fixed target/substrate? Please clarify.
*Here we mean the interaction of light with a solid particle as becomes evident in the following sentences in this section.*

P3, L96. This is only applicable to positive $Cl^+$ ions. Cl ions are readily observed in negative in mode due to high electron affinity.
*We agree and have modified the sentence as follows: "Overall, they observed similar mass spectra in both ns- and fs-laser ablation, but also showed that ions with high ionization energy such as $Cl^+$ are more easily generated by the fs-laser due to its higher power density."*

P3, L199 and P4, L1. Were the beam diameter and focal length/position measured or calculated?

*All ionization laser beams were focused using the same lens. The focal length of the lens was measured for all wavelengths and the beam diameter at focus or interaction position was calculated. This information is now given in sections 2.1., 2.2., and in the supplementary section (cf. Table S4).*

*We have added the following text to section 2.1:*
*"We did vary the laser focus to the left/right and up/down and determined the diameter of the particle beam to 1 - 2 mm, depending on particle type. The ns-laser beam is slightly defocused at the position (F1) increasing the particle-laser interaction area, and the defocused beam diameter is 99±31 μm where it encounters the aerosol particle (F1, Fig. 1). The focus position of the excimer laser is at 20 cm from the lens, and ionization happens 3 - 4 cm after the*

*focus position, for F2 and F1, respectively. This is the distance from focus point to the center of the ion extraction region from where the ions are extracted into the mass analyser. The movable lens can be used to shift the focus position from F1 to F2 where the defocused beam diameter is 81±7 µm resulting in higher power densities acting on the particles. Please note that the position of the ionization region is quite well defined in this case, close to the center of the ion extraction zone, due to the scattering signal of the second detection laser whereas for the experiments with the fs laser we had to apply a different procedure to define this (see section 2.2 and 3.1). Variation of the focus position allows to vary the power density by a factor of ~1.5 for otherwise similar conditions, for F1 and F2, respectively."*

*We have added the following text to section 2.2:*
*"To define the ionization region for this case also close to the center of the ion extraction region a procedure selecting those mass spectra with more than 90% of the maximum total ion intensities was applied (cf. section 3.1)."*

*"For the wavelength of 800 nm the laser beam diameters are 487±77 µm and 246±36 µm at the positions F1 and F2, respectively. The focal positions were varied to study the effect of power density on the mass spectra. The power densities at F2 are ~3.5 times higher than at F1."*

P4, L3. Do you mean the excimer focal position? How do you know ionization happens 3-4cm after the focal position with a counter propagate geometry? Can you comment on the depth of field of the focusing? Is it more likely that ionization takes place before the focal position? Does the position of ionization depend on the absorbing properties and ionization threshold of the material with a counter propagate geometry?

*Yes, we mean the focus position of the excimer laser. The ionization happens typically 2 cm before the focus point of the excimer laser in the counter direction of the particle beam close to the position of the second detection laser. Hence, the ionization happens before the focus point, which improves the hit rate due to the larger laser beam diameter. We have shifted the focus position up to 4 cm away from the ionization region to study the impact on the mass spectra. Please note that the position of the ionization region is quite well defined in this case, close to the center of the ion extraction zone, due to the scattering signal of the second detection laser which triggers the ionization laser beam. Between scattering and ionization a particle travels 500 µm depending on its size, shape, and density. For the experiments with the fs laser we don't have this additional information and applied a different procedure to define this (cf. section 3.1). The position of ionization shouldn't depend strongly on the absorbing properties and ionization threshold of the material as it is defined by the laser triggering (ns-laser) and the dimensions of the ion extraction region (fs-laser).*

P4, L129. What is the diameter of the focal point of the fs system?
*We calculated the laser beam diameters at the focus point and other positions for all wavelengths of the fs and ns-laser system. This calculation and the results are added to the supplementary information of the revised manuscript:*

*Table S4: The Laser beam diameters calculated for the different focus positions and different wavelengths*

| Laser and Wavelength | Beam Diameter at Position $F_2$ (µm) | Beam Diameter at Position $F_1$ (µm) | Beam Diameter at Focus Position (µm) |
|---|---|---|---|
| *fs-laser 800 nm* | *246±36* | *487±77* | *42±9* |
| *fs-laser 266 nm* | *182±32* | *270±32* | *38±9* |
| *ns-laser 193 nm* | *81 ±7* | *99±31* | *37±2* |

P4, L141. Can the authors comment on the stability/reproducibility of the focal length, the depth of field and the effect on power density? How is the power density different at F1 than F2? Surely the densities are the same and the focal position has just shifted upstream in the particle beam. Why do the authors vary the focal length? The objectives and conclusions of this operation are not stated in the manuscript.
*The beam pointing stability is better than 0.25 µrad (rms) for the fs-laser as specified by the manufacturer, and thus also focal length, depth of field, and power density are very stable. The power density is varied by changing the beam diameter of the defocused beam. The focus position was varied to study the variation of power density without*

155 *changing other parameters. We can vary the power density by changing the pulse energy or by changing the focal position. The focus position was changed by changing the lens position from F1 to F2; the beam diameter is larger at position F1 compared to F2 and hence, the power density at F2 is larger than at F1. This is now mentioned in sections 2.1 and, 2.2., and the results are discussed in section 3.2.1.*

P5. L171 and throughout the manuscript. How were the mass spectra assigned to a spectra type?
160 *This was explained above and is detailed now in section 3.1. of the revised manuscript: "The mass spectra that are discussed in this section were selected to be representative for each particle type in the following manner. From the typically 200-600 useful single particle mass spectra measured for each particle type only those 30-40% (60-240 spectra) with at least 90% of the maximum total ion intensities were selected to ensure optimal hit of the particles by the ablation and ionization laser. These remaining spectra were classified using the fuzzy c-mean algorithm available*
165 *in the LAAPTOF data analysis software. This resulted in typically two classes of mass spectra per particle type. For each class of mass spectra we manually selected 10 spectra representing all main characteristics and applied an additional mass axis calibration for each spectrum. These 10 spectra showed correlation coefficients of r = 0.7-0.9. An example demonstrating the reproducibility and representativeness of this selection process is given in the supplementary section (cf. Figures S3-S6)."*

170
P7, L243. Silicate particles predominantly come from mineral dust not sea spray. SiO4 is an anion in crystalline orthosilicates and is not an appropriate description of silicate composition.

175 *We agree; this is an obvious error. We have modified the sentence to: "Recent studies with a LAAPTOF by Marsden et al. (2016) of silicate rich ambient dust particles featured similar mass spectral peaks, namely from $Si^+$, $SiO^+$, and $O^-$, $SiO^-$, $SiO_2^-$ ions."*

P7, L256. The work cited used Silicon substrate with organic and inorganic solutions and the claims regarding the fragmentation/ clusters were relating the compounds in solution not the silicon. Please be careful not to confuse
180 clusters with fragments.

*Kato et al. (2007) have observed more atomization, and cluster formation in fs-laser, ablation and more fragmentation in ns-laser ablation. The clusters (with n=1 to 6) were observed from the silicon substrate and not from the solution.*

185 P10, L361. It should be pointed out that the conclusions are with respect to SPMS with a counter propagate geometry.
*We agree and have modified this sentence. "Generally, ns-laser spectra for the same particle type exhibit higher reproducibility of the spectral pattern than fs-laser spectra with the LAAPTOF in counter propagating geometry".*

P10, L361. Do you mean reproducibility of average ion intensity or spectral pattern?
190 *We mean the reproducibility of spectral pattern. With the ns-LAAPTOF we can observe a higher reproducibility of the spectral pattern.*

P10, L369. Cl- was observed in negative ion spectra. See comment above.

195 *Thanks for this hint: The sentence was modified to: "For NaCl particles, only in fs-laser spectra high ionization energy species like $Cl^+$ or $N^+$ were detected."*

**Technical Corrections**

200 P2, L92. GmbH not Gmbh.
*Corrected.*

P10, L372. Sentence structure/word order.
*For NaCl particles, only in fs-laser spectra high ionization energy species like $Cl^+$ or $N^+$ were detected. Fs-laser*
205 *ablation also led to formation of oxides, for e.g. core-shell particles, silica particles, and silver oxides for the gold-silver core-shell particles.*

Figure1 Inset. It is not clear what the position label, defocus label and arrow are referring to.
*The inset picture is modified and labels are marked clearly.*

210

Figure 2. It is not clear what the wavy line represents.

*It is an input data signal, and is marked on the schematic diagram. This diagram explains the pre-trigger data acquisition technique. The figure caption has been modified to: "Figure 1: Pre-trigger sampling mechanism of the data acquisition system. The data (input signal, the dotted wavy line) arriving before the trigger event (pre-trigger samples) are saved in the temporary memory of the data acquisition card and then combined with the trigger samples"*

215

Figures 3-9. Please state the estimated power density or laser setting at which these spectra were acquired.

*The pulse energies and power densities were added to each figure caption.*

220

Figures 9 and 10. Please state how many particles were averaged and what the error bars represent. The error bars are difficult to see and the quality of the graphics are generally poor.

*The averaged intensities are calculated for 10 representative spectra of each particle type and at each power density. The error bars are standard deviations of the average intensity. Both the graphs Figure 9 and 10 were modified for better visibility. The caption of Figure 10 was modified to: "Figure 10: Variation of average total ion intensity with respect to the size of PSL particles. The total ion intensities were averaged for 10 representative spectra of each particle type. (Particle diameters: 500 nm, red circles, and 1000 nm, green triangles). (a, b) at focus position F2 and (c, d) at focus position F1."*

225

---

## Author Comment (AC2) · 23 Mar 2018

**Reply to Reviewer 2**

*We gratefully thank the reviewer for the careful manuscript reading, and the constructive comments which were helpful to improve the quality of our manuscript. Our point-to-point replies are given below in italics and in blue following the original comments:*

General comments

This paper deals with the single particle mass spectrometry (SPMS) of atmospherically relevant aerosol particles (or laboratory surrogates) in the one-step (ablation/ionization) laser approach, comparing the performances of ns (193 nm) and fs (800 nm and 266 nm) laser irradiation. As a preliminary comment, although this technique can be useful for classifying atmospheric particles based on their (fragmentation) mass spectral fingerprints, it does not provide detailed molecular information on the actual chemical composition of the particles. The two-step (separate ablation and ionization lasers) approach is much more effective on this (see, e.g., Zimmermann's group studies, Anal. Chem. 2017, 89, 6341). The introduction should better acknowledge this, i.e. mention the actual chemical analysis capabilities, not only the improvement of quantitative abilities in SPMS.

*We have added the following sentence to the introduction: "Although a two-step approach separating laser ablation and laser ionization bears several advantages e.g. to identify specific molecules (Passig et al., 2017) currently still many instruments use a single step laser desorption and ionization."*

The paper presents an important amount of experimental data. However, it is written mostly at a descriptive and
comparative level, with no deeper insight into the actual mechanisms involved in the ablation and ionization processes. Purely speculative assertions (e.g. ". . . which again indicates a more complex ionization mechanism during fs-laser ablation", rows 212-213, or rows 179-182, 334-336 etc.), cannot contribute much to advancing our knowledge on this technique.

*In our paper we discuss observed differences between mass spectra from using fs laser light and ns-laser light interacting with aerosol particles. We base the discussion of our results on current literature on light particle interactions wherever possible (Amoruso et al., 1999; Bäuerle, 2011; Zaidi et al., 2015; Zhou et al., 2006). However, most studies on light-matter interactions used solid substrates, and mechanisms are thus not easily comparable with this work, with a few exceptions (Zawadowicz et al., 2015). The principle processes governing*
*interaction of fs-laser light with single particles remains subject to future investigations. However, we have tried to formulate the paragraphs mentioned by the reviewer in a better way:*

179-182:
"One explanation for this observation could be that the type 2 spectra are generated from particles that are ionized
closer to the positive extraction region, whereas the type 1 spectra may arise from particles ionized closer to the negative extraction region or in the middle of the ionization region of the mass spectrometer."
 *The section 3.1.1 is updated with the following lines.*

*"One explanation for this observation could be that the type 2 spectra are generated from particles that are ionized closer to the positive ion extraction region, whereas the type 1 spectra may arise from particles ionized*
*closer to the negative ion extraction region or in the middle of the ion extraction region of the mass spectrometer. Since the particle beam at the ionization region has a width of 1-2 mm and the laser beam a width between 487±77μm (F1) and 246±36 μm (F2) it is possible that some particle are ionized closer to either one of the electrodes leading to these two types of mass spectra."*

212-213
"We also observed more cluster ions in the fs-laser spectra ($Na_2Cl^+$, $NaCl_2^+$, $Na_2Cl^-$, $Na_2Cl_3^-$ and $Na_4Cl_4^-$) compared to the ns-laser spectra, which again indicates a more complex ionization mechanism during fs-laser ablation."
*The following lines are added to the section 3.1.2.*

*"Several studies on fs-laser ablation of NaCl have observed the formation of cluster ions at higher power densities due to Coulomb or phase explosion, depending on excitation energy (Hada et al., 2014; Henyk et al., 2000a, b; Reif et al., 2004)."*

334-336
      *"The observed slight saturation effect of signal intensity at higher power densities for both lasers may be due the coulombic repulsion among the ions during multiphoton ionization, observed as well by L'Huillier et al. (1987)."*
      *The section 3.2.1 is modified as follows:*

*"Based on our limited data and the available literature one can only speculate about potential reasons. The observed slight saturation effect of signal intensity at higher power densities for both lasers and most particle types may be due the Coulomb repulsion among the ions during multiphoton ionization, observed as well by L'Huillier et al. (1987). Furthermore, the penetration of the plasma into the particles with increasing power density may be limited e.g. due to absorption of part of the additional power by the plasma near the surface."*

Moreover, as the authors acknowledge themselves, this paper is an extension of a previous one (Zawadowicz et al., Anal. Chem., 2015), the difference being an "in-line" laser irradiation, compared to the orthogonal one used in the previous paper. The choice of this new configuration is not justified by the authors. It introduces an important experimental uncertainty on the actual ablation/ionization position in the ion source of the bipolar mass spectrometer (evaluated by the authors at 2-4 cm from the focus – how was this calculated?), which further generates a lack of precision in the discussion (see, e.g., rows 179-182). Additionally, this results into significant uncertainty on the laser irradiance actually experienced by the particles at the ablation/ionization spot. The reported laser irradiances, calculated in the focal plane, are therefore mostly useless for comparison with experiments performed by other groups in different geometries.

*We have pointed out in the introduction why we used the collinear setup for laser light particle interaction: "Please note that for this work the geometry of ablation/ionization laser beam particle interaction was not orthogonal as for the experiments described by Zawadowicz et al. (2015), but almost collinear as this was favoured by the design of the LAAPTOF."*

*Our description of the potential positon for fs-laser light particle interaction was indeed misleading, such that the reader did get the impression there would be an uncertainty of 2-4 cm. This is however not the case, due to the procedure we apply to select the spectra used for analysis. This is now described in more detail in section 2.1 for the ns-laser:*

*"We did vary the laser focus to the left/right and up/down and determined the diameter of the particle beam to 1 - 2 mm, depending on particle type. The ns-laser beam is slightly defocused at the position (F1) increasing the particle-laser interaction area, and the defocused beam diameter is 99±31 µm where it encounters the aerosol particle (F1, Fig. 1). The focus position of the excimer laser is at 20 cm from the lens, and ionization happens 3 - 4 cm after the focus position, for F2 and F1, respectively. This is the distance from focus point to the centre of the*

*ion extraction region from where the ions are extracted into the mass analyser. The movable lens can be used to shift the focus position from F1 to F2 where the defocused beam diameter is 81±7 µm resulting in higher power densities acting on the particles. Please note that the position of the ionization region is quite well defined in this case, close to the centre of the ion extraction zone, due to the scattering signal of the second detection laser whereas for the experiments with the fs laser we had to apply a different procedure to define this (see section 2.2*

*and 3.1). Variation of the focus position allows to vary the power density by a factor of ~1.5 for otherwise similar conditions, for F1 and F2, respectively."*

      *And in section 2.2 for the fs-laser:*

      *"To define the ionization region for this case also close to the centre of the ion extraction region a procedure*
*selecting those mass spectra with more than 90% of the maximum total ion intensities was applied (cf. section 3.1). A movable focusing lens set-up was used for multiple focusing positions between F1 and F2 further towards inlet, to better understand the effect of power density on mass spectral patterns (insert in Fig. 1). The laser beam diameters are calculated and for all three wavelengths and for two different focus positions (cf. Table S4). For the wavelength of 800 nm the laser beam diameters are 487±77 µm and 246±36 µm at the*
*positions F1 and F2, respectively. The focal positions were varied to study the effect of power density on the mass spectra. The power densities at F2 are ~3.5 times higher than at F1."*

*And in section 3.1 for the fs-laser:*
*"The mass spectra that are discussed in this section were selected to be representative for each particle type in the following manner. From the typically 200-600 useful single particle mass spectra measured for each particle*
*type only those 30-40% (60-240 spectra) with at least 90% of the maximum total ion intensities were selected to ensure optimal hit of the particles by the ablation and ionization laser. These remaining spectra were classified using the fuzzy c-mean algorithm available in the LAAPTOF data analysis software. This resulted in typically two classes of mass spectra per particle type. For each class of mass spectra we manually selected 10 spectra representing all main characteristics and applied an additional mass axis calibration for each spectrum. These*
*10 spectra showed correlation coefficients of r = 0.7-0.9. An example demonstrating the reproducibility and representativeness of this selection process is given in the supplementary section (cf. Figures S3-S6)."*

The conclusion of the paper is that the use of a fs laser presents rather limited interest, when compared to much common (and cheaper) ns sources. This conclusion can be a bit rushed. Although not evident here, the fs approach
can still have an interesting application in depth profiling of phase-separated or mixed aerosol particles, but for this a precise and reproducible alignment of the laser beam with respect to the particle must be achieved, which is clearly not the path followed in this study.

*We also thought that the fs-laser application could have the potential advantages you mentioned but could not*
*substantiate this experimentally with the existing technical setup of the fs-LAAPTOF and e.g. the core-shell reference particles. However, we think that we used in this study a precise and reproducible alignment of the laser beam with respect to the particles as discussed above.*

*The following lines are added to the conclusion section:*
*"The idea that the higher power density on the particles which can be achieved with fs-laser pulses leads to a more complete ablation and ionization could not be substantiated in this study. However, the cluster formation nature of fs-laser ablation rewards more studies with aerosol particles to understand and correlate the results for potential improvements in quantification and mixing state analysis. Further tests including e.g. two step ionization or delayed extraction are needed to investigate potential advantages of fs- over ns-laser ablation in atmospheric*
*SPMS."*

It is very surprising that the two wavelengths (800 nm and 266 nm) used for fs ablation/ionization returned absolutely similar results. On one hand, the multi-photon ionization (MPI) invoked by the authors is very different at the two wavelengths (three times more photons needed in IR to reach the same ionization energy), this should
result in orders of magnitude difference in the MPI yield, which could not be compensated by the 20-fold higher IR energy/pulse. On the other hand, the optical properties of the studied particles (although extensively mentioned in the Introduction) are not properly used in the text to account for the experimental observations. Most of the discussion (e.g. high reflectance of Au at 800 nm) is based on single-photon interaction assumptions, while at the high intensities reached in fs regime everything is so multi-photonic (i.e. non-linear). Moreover, the optical
properties at 266 nm are completely ignored.

*We also expected to observe larger differences in the mass spectra when using fs-laser pulses of 800 and 266 nm. However, firstly, the lower energy (0.2 mJ) of the 266 nm UV-fs-laser pulses has led to less usable spectra due to reduced light scattering and corresponding trigger signals. Hence, the discussion of the resulting mass spectra is*
*based on only a small number of spectra which leads to a larger uncertainty. Secondly, we have high power densities for either fs-laser wavelength and hence multi-photon ionization followed by Coulomb and/or phase explosion leading to similar ions. Even the substantial difference in reflectivity e.g. of the core-shell particles for the two different wavelengths didn't cause any significant change in the mass spectra. To discuss the findings for fs-laser ablation at 266 nm a new section 3.1.7 was added with the following text:*

*The following sub-section is added as new section 3.1.7.*
*"The mass spectra obtained for fs-laser pulses of 266 nm wavelength and 0.2 mJ energy/pulse show very similar features for all the samples measured as obtained for the other fs-laser wavelength of 800 nm. Please note that about 80% of the spectra collected for 266 nm were empty due to reduced light scattering signal and*
*corresponding ineffective triggering of mass spectra recording. Furthermore, the mass spectra containing information have a 3 to 5 times lower intensity for all particle types compared to those obtained for fs-laser pulses of 800 nm, with a similar energy of 0.3 mJ per pulse. However, the spectral features are similar for all samples.*

*This is shown for PSL particles of 500 nm diameter in Figure 3.c and Figure 3.d.with the only remarkable difference being the reduced ion intensity. In the case of NaCl, the UV fs-LAAPTOF results in the same positive*

*spectra as for 800 nm (cf. Figure 04.c) and only major Cl- peaks in the negative spectra. Na2Cl+, NaCl2+ also exist in the positive spectrum. However, the ion intensity is 4 times smaller than with 800 nm. The ammonium nitrate particles have the same positive (NH2+/O+, OH+, NH4+, NO22+, NO+, and NO2+ ) and negative ion spectral features as shown in Figure 07.c-d (800 nm) but with less intensity. The SiO2 particles show almost exactly the same spectral features in the negative ion spectra for both wavelengths. Also the major positive ions*

*O+,Si+,O2+, SixOy+ (x = 1-3, y = 2x+1) are found for both wavelengths and remaining peaks have much less intensity. A single particle spectrum for a SiO2 particle is shown in Figure S.7. For all core- shell particles, the spectral signatures originating from additional surface coatings e.g. by water or the surfactant (Cetyl-trimethylammonium bromide) are nearly the same for 266 and 800 nm. No gold signature was observed for any of the core-shell particles using 266 nm and 800 nm fs-laser pulses despite the lower reflectivity of gold in the*

*UV."*

*The following lines are added in the conclusion section 4.*
*"However, for fs-laser ablation it seems that the rapid plasma formation on the surface e.g. of the core-shell particles prevents deeper impact and hence ablation and ionization of core material at least for shell thicknesses*

*of 150 nm. The mass spectra available from the fs-laser with 266 nm and an energy of 0.2 mJ have shown very similar spectra as for the fs-laser operating with 800 nm and 0.3 mJ. Despite the relative small number of usable spectra for 266 nm we consider it very likely that high power densities and hence multi-photon ionization taking place for both wavelengths lead to the formation of similar ions which points to similar ion formation mechanisms. However, a more detailed discussion of possible ion formation mechanisms is not possible based on the data*

*available."*

Specific comments

1.     Rows 68-69: "The energy per unit volume is greater for femtosecond laser pulses compared to nanosecond laser pulses" – why? The energy/pulse is comparable for ns and fs (800 nm). Is the focusing different? Can you clearly specify the beam diameter at the focus (or better, at the interaction) spot for all three beams used? For each experiment: please indicate clearly the energy per pulse used.

*The energy per pulse is indeed comparable with 4 mJ for the ns –laser and 3.2 mJ for fs-laser (800 nm). We have included these numbers in the description of each type of particles, and in the caption of each*

*mass spectrum. The beam diameters for each laser wavelength and position were calculated and tabulated in the supplementary section (Table S4).*

*Table S4: The laser beam diameters calculated for different positions*

[revised manuscript text omitted]

3.    Rows 120-122: "beam diameter is ~300 μm" – how was this value obtained? Calculated/measured? If calculated, were the lens aberrations taken into account? If the same lens was used for 193 nm and 800 nm, how the focal length and all subsequent calculations change between these two wavelengths? What is the error bar on the beam diameter? Error bars should be indicated also on the irradiance values all over the manuscript (including Figures). How was the 2-4 cm position after the focus determined? Does this translates into 1-3 cm for F1 focusing? How these values change between the three wavelengths, considering the change in the focal length? F1 and F2 mentioned here are not indicated in Figure 1. Please give the beam diameter limits in the (2-4 (1-3) cm?) laser-particle interaction region for all wavelengths, this would be much more useful than diameter at the focus. Indicate also the irradiance limits with the error bars related to calculations and measurements.

*We used the same lens of 20 cm focal length to focus the ns as well as the fs laser beams. The laser beam diameters were calculated at the positions F1 and F2, respectively. The lens aberrations were taken into account for this calculation and also for the alignment of the optical set up. The uncertainties given for the beam diameter at interaction position are considered for the variation in the pulse energy and the*

*spherical aberration. However, the impact of the uncertainties in laser beam diameter on power densities ranges between 18 and 36% and is hence relatively small compared to the overall variation of the power densities.*

*The distance 3- 4 cm from the focus point is given by the ion extraction region of the mass spectrometer. The ions cannot be extracted into the mass analyzer if the ionization happens outside this region. The*

*focal position was changed from F1 to F2 to vary the power density by moving the lens L towards center of the ionization region. The positions of F1 and F2 are updated in Figure 1. We have added more details on this in the manuscript in the sections 2.1, and 2.2. Please refer also to the answers we have given to the comments of reviewer 1 related to this topic.*

*Section 2.1:*

*"We did vary the laser focus to the left/right and up/down and determined the diameter of the particle beam to 1 - 2 mm, depending on particle type. The ns-laser beam is slightly defocused at the position (F1) increasing the particle-laser interaction area, and the defocused beam diameter is 99±31 μm where it encounters the aerosol particle (F1, Fig. 1). The focus position of the excimer laser is at 20 cm from*

*the lens, and ionization happens 3 - 4 cm after the focus position, for F2 and F1, respectively. This is the distance from focus point to the centre of the ion extraction region from where the ions are extracted into the mass analyser. The movable lens can be used to shift the focus position from F1 to F2 where the defocused beam diameter is 81±7 μm resulting in higher power densities acting on the particles. Please note that the position of the ionization region is quite well defined in this case, close to the centre of the*

*ion extraction zone, due to the scattering signal of the second detection laser whereas for the experiments with the fs laser we had to apply a different procedure to define this (see section 2.2 and 3.1). Variation of the focus position allows to vary the power density by a factor of ~1.5 for otherwise similar conditions, for F1 and F2, respectively."*

*Section 2.2.:*

*"A movable focusing lens set-up was used for multiple focusing positions between F1 and F2 further towards inlet, to better understand the effect of power density on mass spectral patterns (insert in Fig. 1). The laser beam diameters are calculated for all three wavelengths and for two different focus positions (cf. Table S4). For the wavelength of 800 nm the laser beam*

*diameters are 487±77 μm and 246±36 μm at the positions F1 and F2, respectively. The focal positions were varied to study the effect of power density on the mass spectra. The power densities at F2 are ~3.5 times higher than at F1."*

4. Please report mass resolution for both polarities. From Figures, this seems to be around 100. In these

conditions, how certain can be the assignment of some mass comment peaks, e.g. m/z 16, 18?

*The mass resolution of the mass spectrometer is given by the manufacturer Tofwerk AG as m/Δm =600-800 for m/z = 1-2000. We observed resolutions between m/Δm =300-700 at full width half maxima for both polarities. Analysis of mass spectra from this work resulted in mass resolutions for masses 16, 24*

*and 48 of 458, 530, and 593, respectively. At this resolution we can distinguish peak differences on a single mass unit basis. Please note that most difficulties in peak assignment don't originate from mass resolution, but from the jitter of the mass axis from spectrum to spectrum or particle to particle. The information on the resolution of the mass spectrometer is now given in section 3.1;*

*"Analysis of mass spectra for both polarities from this work resulted in mass resolutions at full width half maxima for masses 16, 24 and 48 of 458, 530, and 593, respectively. At this resolution we can distinguish peak differences on a single mass unit basis. Please note that most difficulties in peak assignment don't originate from mass resolution, but from the jitter of the mass axis from spectrum to spectrum or particle to particle."*

5. Rows 179-182: the explanation for observation of type 1 vs type 2 spectra is not convincing. Can the authors provide a more developed explanation, based on experimental evidence? Generally speaking, a more thorough discussion on type 1 vs type 2 spectra is needed (see also comment 12 below), as this can have practical implications on particle classification in "real world" (field) experiments.

*While optimizing the position of the ns ionization laser we observed a loss of negative ion signal if the ionization laser was closer to the positive extraction electrodes, and vice-versa. Since the particle beam at the ionization region has a width of 1-2 mm, and the laser beam a width of $246\pm36\mu m$ (F2) and $487\pm77\mu m$ (F1), it is possible that some particles are ionized closer to either one of the electrodes, which*

*may result in these two types of mass spectra.*

*We have added the following sentence to section 3.1.1.:*

*"One explanation for this observation could be that the type 2 spectra are generated from particles that are ionized closer to the positive ion extraction region, whereas the type 1 spectra may arise from*

*particles ionized closer to the negative ion extraction region or in the middle of the ion extraction region of the mass spectrometer. Since the particle beam at the ionization region has a width of 1-2 mm and the laser beam a width between $487\pm77\mu m$ (F1) and $246\pm36$ $\mu m$ (F2) it is possible that some particle are ionized closer to either one of the electrodes leading to these two types of mass spectra."*

6. Rows 183-188: formation of larger carbon clusters for fs-ablation: "This may be due to the higher power

density of the fs-laser, and reactions of the primary ion species with the source plume forming larger clusters as secondary products" – what is the experimental evidence for the in-plume growth of these clusters? How their intensity changes with the increase in laser irradiance? Please show the data (at least in Supplementary Information), they must be available from studies performed in section 3.2. In the Conclusion sections, the in-plume reactions are not mentioned, but only formation at the ablation stage

(rows 366-367). Please put in agreement the conclusions with the main text assertions.

*We have generally observed larger ion clusters for fs-laser ablation than for ns-laser ablation and for increasing power density. We have added an example of how the cluster intensities change with increasing laser irradiance in the supplementary information (Figure S7). The following sentences were added to the results and conclusion sections:*

*Section 3.1.1: "In both laser ablation methods we observe formation of carbon clusters and hydrogenated carbon cluster ions from PSL particles. For fs-laser ablation, larger carbon clusters (> 7 carbon atoms) with (in positive mode) fewer hydrogen atoms (< 3 hydrogen atoms) are observed. Such larger clusters in the fs-laser spectra can potentially form during the Coulomb or phase explosion of the fs-laser ablation process but some studies claim that also reactions of the primary ion species with the source plume can*

*generate the larger clusters (Zaidi et al., 2010). For both laser pulse durations, the number of larger clusters increased with increasing laser pulse energy for the PSL spectra as has also been reported for ns-laser pulses by Weiss et al., 1997."*

*Section 3.1.4: "The increasing abundance of larger clusters with increasing laser pulse energy is shown in Figure S7 for $SiO_2$ particles."*

*Section 4: "Such larger clusters in the fs-laser spectra can potentially form during the Coulomb or phase explosion of the fs-laser ablation process. Some studies claim that also reactions of the primary ion species with the source plume may generate the larger clusters (Zaidi et al., 2010). However, these complex processes of fs-laser ionization are beyond the scope of this paper but require further studies."*

7. Section 3.1.2: the optical properties of NaCl particles are quite well-known and should be used to explain the low efficiency in generating mass spectra in fs AND ns regimes.

*The hit rate (mass spectra produced from particles) is smaller in the case of NaCl compared to $NH_4NO_3$ particles. This may partially be due to the cubic nature of the particles, which can lead to a wider particle beam resulting in more empty spectra. The weaker light absorption of NaCl may also explain part of the*

*observation. Generally, the salts absorb more light at 193 nm compared to 800 nm which could explain*

*some of the difference between ns and fs results. Hence, both, morphology and optical properties can have an impact on the hit rates observed.*

*The following sentence was added to section 3.1.2: "This low hit rate for the fs-laser compared to the ns-laser may be related to both, the particle shape widening the particle beam and the reduced absorption at 800 nm compared to 193 nm."*

8.   Rows 211-213: please clarify what you mean. Are these species generated in the ablation process, or by subsequent interactions in the plume? What is the role of the ionization here?

*Formation of new clusters after ablation or photoionization was observed by Henyk et al. (2007), Bulgakov et al., and Zaidi et al. (2010) for the fs-laser ablation/ionization of NaCl, BaF$_2$, Si, and methane and in several other studies on silicon clusters (Bulgakov et al., 2004; Henyk et al., 2000a; P.A. Márquez Aguilar, 2007; Reif et al., 2004). However, the interaction of laser induced plasma with a solid substrate or a solution in these studies is most likely not in all aspects comparable to the single particle laser ablation. NaCl is ablated with the fs-laser leading to atomization (Na$^+$ and Cl$^-$) and cluster ion formation*

*in the Coulomb or phase explosion of the ionization.*

*The following lines are added in the section 3.1.2: "Several studies on fs-laser ablation of NaCl have observed the formation of cluster ions at higher power densities due to Coulomb or phase explosion, depending on excitation energy (Hada et al., 2014; Henyk et al., 2000a, b; Reif et al., 2004)."*

9.   Section 3.1.3: an explanation should be advanced for the very low efficiency in generating mass spectra in both positive (10%) and negative (1%) polarities with the (800nm?) fs laser, with respect to the much higher (100%?) efficiency achieved with the ns one.

*Normal operation of the LAAPTOF with the Excimer laser leads to hit rates above 90% if triggered to the second detection laser. Using the fs-laser in a free firing mode leads to a much lower hit rate (empty spectra), and also a large number of incomplete hits (non-representative low intensity spectra). Consequently, empty and non-representative low intensity spectra were omitted for further analysis. This*

*is explained now in section 3.1 of the revised manuscript.*

10.   Section 3.1.4: less intense signal at 266 nm compared to 800 nm – please try to relate this to the optical properties of the SiO2 particles.

*The refractive index of fused silica is 1.4533 at 800 nm and 1.4997 at 266 (Malitson, 1965). Hence, the optical properties are not that different for both wavelengths. However, one would expect a slightly stronger absorption at 266 nm especially considering potential impurities in the particles. The ion formation mechanism seems to be insensitive to the differences in optical properties for these two wavelengths.*

11.   Rows 290-292: "The high reflectance of gold in the IR likely leads to reduced ablation of the core" – beside the fact that the absorption processes in the fs regime must be highly multi-photonic, this conclusion is questionable, as similar spectra are observed for 266 nm fs irradiation, or at this wavelength the reflectance of gold and silver is much reduced (~30%). How can the authors interpret this? The same explanation is given rows 311-312, although the same similarity is observed between 800 nm and 266 nm irradiation.

*Indeed, there is a significant difference in the reflectance of gold for 266 nm and 800 nm. While the high reflectance of gold and silver in the IR likely contributes to reduced ablation of the core the lower pulse*

*energy at 266 nm may neither be sufficient to produce a significant signal from the core.*

*The following sentence was added to section 3.1.5:*
*"The high reflectance of gold and silver in the IR likely contributes to the reduced ablation of the core. Although the reflectance of these particles is much lower at 266 nm these fs-laser pulses were also not*

*capable to generate a significant signal from the core for the reduced pulse energy of 0.2 mJ."*

12.     Rows 305-309: an explanation for the existence of two types of spectra must be advanced.

*The following lines are added to section 3.1.6 to explain these observations:*

*"The spectra with both organic shell and gold core signature are most likely produced from particles hit very close to the centre of the laser beam. The spectra without gold signature are mot likely produced from particles interacting only with part of the laser beam. Please note that the particle beam has a diameter ranging between 1-2 mm while the laser beam diameter ranges between 246 and 487 µm (cf. Table S4)."*

13.     Row 329: is this average factor relevant, considering the huge variability in efficiently generating usable mass spectra?

*As outlined in the comments to reviewer one and shown by additional material in the supplementary section (Figure S3-S6), we used a well-defined selection procedure for the mass spectra generated by the fs-laser to choose 10 mass spectra we consider as representative and comparable to those generated by the ns-laser.*

14.     Rows 334-336: "... saturation effect ... may be due to coulombic repulsion ..." – please develop. Why
this effect would occur only for NaCl particles? How this saturation correlates with the low efficiency (16%) in generating non-empty mass spectra with the fs laser? How is this saturation effect related to the optical properties of the NaCl particles (vs the others)?

*The low hit rate (lower mass spectra production) with the fs-laser is caused by different reasons as*
*outlined above (response to comment 7). The saturation effect depends on the total number of ions generated, and hence on the ionization efficiency compared to others.*

*The following text was added in the section 3.2.1*

*"Based on our limited data and the available literature one can only speculate about potential reasons.*
*The observed slight saturation effect of signal intensity at higher power densities for both lasers and most particle types may be due the Coulomb repulsion among the ions during multiphoton ionization, observed as well by L'Huillier et al. (1987). Furthermore, the penetration of the plasma into the particles with increasing power density may be limited e.g. due to absorption of part of the additional power by the plasma near the surface."*

15.     Row 345: factor 7 claimed is not evident from Fig. 10, please check.

*Thank you for pointing this out. Indeed the maximum difference in total ion intensity is a factor of 4, and*
*hence a factor of 2 lower than the increase in mass (factor 8). The corresponding changes in section 3.2.2 are given below.*

16.     Row 348: Factor 8 in volume is not "much larger" than factor 7 in ion intensity (if confirmed).

*See response to comment 15, the factor of 8 is much larger than the factor of 2-4. Section 3.2.2.has been modified as follows:*

*"To explore the quantitative abilities of the fs- and ns-laser we also investigated the average ion signal intensity variation as a function of laser power density with respect to particle size (subplots a and b in*
*Fig. 10), using PSL particles of 500 and 1000 nm diameter. Similar subplots (Fig. 10c – d) are shown for focus position F1 with lower power density. The average signal intensity for the 1000 nm size particles as a function of the excimer laser power density is 2-4 times higher compared to the signal intensity for 500 nm particles for both focus positions. The femtosecond laser produced only 1.5 -2 times*

*larger average ion signals for 1000 nm particles compared to 500 nm particles. However, this difference between ns- and fs-laser ion intensities for these different particle sizes is within the uncertainties and also has to be verified for different types of particles. The mass ratio of the two particle sizes is 8, hence much larger than the relative differences in the total ion intensities. The ratio of the surface area of the 1000 nm PSL and 500 nm PSL particles is 4 which is comparable to the maximum intensity difference observed. This could be an indication that the ionization scales with the particle surface area. The increase in ion signal thus does not scale linearly with the difference in mass of the two particles sizes and of the total material potentially to be ablated. Similar effects were observed for RbNO$_3$ and (NH$_4$)$_2$SO$_4$ particles (Reents et al., 1994). This demonstrates the quantitative limitations of both ns- and fs-laser ablation.”*

17. Rows 350-351: “This demonstrates the quantitative limitations of both ns- and fs- laser ablation”. However, can the authors infer something about the fraction of particle mass which is vaporized from the measured data in Figure 10?

*We can't give the absolute fraction of particle mass vaporized since we don't have a reference point for which we would know this fraction or can be sure that the complete particle has been vaporized.*

**Technical corrections**

18. Rows 80-81: ablated particles cannot move 5 μm during 5 ns and 0.1 μm during 100 fs, please check

*This estimate is based on ablated ions average speed of 1000 m/s caused by acceleration into the vacuum and is therefore the maximum distance that can be covered by the ablated remnants (Marine et al., 1992; Walsh and Deutsch, 1991).*

*The following sentence was added to the introduction:*

*“Please note, the ablated particle components move up to ~5 μm during a 5 ns pulse or ~0.1 μm during a 100 fs pulse even, respectively, and in both cases remain well within the typical laser beam width. This estimate is based on an average ion speed of 1000 m s$^{-1}$ (Marine et al., 1992; Walsh and Deutsch, 1991).”*

19.  Rows 301-303: 44% + 66% = 110%
*We corrected 66% to 56% in the text.*

20. Rows 387-388: please check English
*Corrected.*

21. Tables 1 and 3 can go to Supplementary Information
*We moved Table 3 to the SI, but consider Tables 1 and 2 together as useful for the method section.*

22.  Table 2 is useless in this form, everyone can apply the proportionality on the energy/pulse. Give instead proper beam diameters in the interaction zone for each wavelength (see above)

*This is a good suggestion. We have added this information in Table S4.*

23.  Fig. 10 caption: inversion red-green
*This is corrected and the figure is updated for better visibility.*

Amoruso, S., Bruzzese, R., Spinelli, N., and Velotta, R.: Characterization of laser-ablation plasmas, Journal of Physics B: Atomic, Molecular and Optical Physics, 32, R131, 1999.

Bäuerle, D.: Ultrashort-Pulse Laser Ablation. In: Laser Processing and Chemistry, Springer Berlin Heidelberg, Berlin, Heidelberg, 2011.

[revised manuscript text omitted]